# Biphasic activation of β-arrestin 1 upon interaction with a GPCR revealed by methyl-TROSY NMR

Yutaro Shiraishi [1], Yutaka Kofuku[2], Takumi Ueda [2], Shubhi Pandey[3], Hemlata Dwivedi-Agnihotri[3], Arun K. Shukla [3] & Ichio Shimada [1] ✉

β-arrestins (βarrs) play multifaceted roles in the function of G protein-coupled receptors (GPCRs). βarrs typically interact with phosphorylated C-terminal tail (C tail) and trans-membrane core (TM core) of GPCRs. However, the effects of the C tail- and TM core-mediated interactions on the conformational activation of βarrs have remained elusive. Here, we show the conformational changes for βarr activation upon the C tail- and TM core-mediated interactions with a prototypical GPCR by nuclear magnetic resonance (NMR) spectroscopy. Our NMR analyses demonstrated that while the C tail-mediated interaction alone induces partial activation, in which βarr exists in equilibrium between basal and activated conformations, the TM core- and the C tail-mediated interactions together completely shift the equilibrium toward the activated conformation. The conformation-selective antibody, Fab30, promotes partially activated βarr into the activated-like conformation. This plasticity of βarr conformation in complex with GPCRs engaged in different binding modes may explain the multifunctionality of βarrs.

[1] Laboratory for Dynamic Structure of Biomolecules, RIKEN Center for Biosystems Dynamics Research (BDR), 1-7-22 Suehiro-cho, Tsurumi-ku, Yokohama, Kanagawa 230-0045, Japan. [2] Graduate School of Pharmaceutical Sciences, The University of Tokyo, Hongo 7-3-1, Bunkyo-ku, Tokyo 113-0033, Japan. [3] Department of Biological Sciences and Bioengineering, Indian Institute of Technology, Kanpur 208016, India. ✉email: ichio.shimada@riken.jp

After activation by agonist binding, G protein-coupled receptors (GPCRs) are typically phosphorylated at serine and/or threonine residues on their C-terminal region by GPCR kinases (GRKs), and β-arrestins (βarrs) bind to the phosphorylated GPCRs, leading to signal regulation via GPCRs. βarr binding to GPCRs elicits multifaceted responses such as receptor desensitization, receptor internalization, and activation of mitogen-activated protein kinases (MAPKs). The functions of βarrs are driven by scaffolds of various downstream effectors, such as clathrin, adaptin, kinases, and phosphatases. The ability of GPCR-bound βarr to engage different scaffolding partners suggests that βarr can adopt multiple conformations, which are modulated by GPCRs[1]. Recent reports suggested that the multiple functions of βarr, such as desensitization, internalization, and signal transduction, could be separable by mutations of βarrs or ligands bound to GPCRs, and different readouts of βarr functions resulted in distinct pharmacological outcomes[2–5]. Thus, the interactions between GPCRs and βarrs are emerging as a focal point for drug discovery, and the details of the conformational states of βarrs bound to GPCRs should be clarified to provide information for developing functionally selective drugs.

βarrs are composed of N and C lobes, and each forms a β-strand sandwich structure. The N and C lobes are connected by a hinge region, and the finger loop is located near the interface between the N and C lobes[6]. βarrs bind to two distinct elements in GPCRs: the carboxyl terminal region phosphorylated by GRKs (also referred to as "C tail"), and the cytoplasmic face of the agonist-activated transmembrane region (also referred to as "TM core")[7]. Structural analyses revealed that the finger loop and N lobe of βarrs bind to the TM core and C tail of GPCRs, respectively[8–12]. Due to the biphasic nature of the interaction, the effects of the C tail- and TM core-mediated interactions on the conformation of βarrs should be separately investigated to understand their specific conformational activations. Growing evidence has suggested that the C tail interaction alone can induce signal transduction at least for some pathways[2,4]. Furthermore, recent studies have indicated that the TM core can activate βarr without the C tail-mediated interaction[13,14]. These findings suggest that the C tail- and TM core-mediated interactions could induce the conformational changes of βarrs. However, the effects of these interactions on the conformational changes for βarr activation in solution have remained elusive, because activated arrestin is inherently dynamic[15] and the available structures have been visualized with stabilization methods, including pre-activated mutants[8,10,12], fusion proteins with a GPCR[8,12], and the conformation-selective antibody fragment, Fab30[9,11,12].

Nuclear magnetic resonance (NMR) spectroscopy is a promising approach for analyzing relationships between dynamic structural information and functions of biomolecules in solution. Here, we investigated the conformational changes of βarr1 induced by C tail- and TM core-mediated interactions with GPCRs by methyl transverse relaxation-optimized spectroscopy (TROSY). We also assessed the effect of the synthetic antibody Fab30 on the conformation of βarr1. Our results suggest that both the C tail- and TM core-mediated interactions with GPCRs are required for the full conformational changes for βarr1 activation in solution, and that Fab30 promotes the conformational changes for βarr1 activation without the TM core-mediated interactions with GPCRs. This conformational plasticity of βarr1 in complex with a GPCR may explain its ability to engage multiple scaffolding partners and in turn mediate distinct functional outcomes.

## Results

### Phosphorylation- and agonist-dependent interactions between βarr and GPCRs.
To prevent intra- and intermolecular disulfide bond formation and enhance the stability of βarr1, we used its cysteine-less variant, which showed similar receptor-binding activity to wild-type βarr1[16]. We also used a modified β2 adrenergic receptor (β2AR) construct with its C-terminus replaced by that of arginine-vasopressin type2 receptor (V2R), referred to as β2V2R. This receptor construct reportedly has higher affinity for βarr1 than wild-type β2AR, and thus was used in previous structural analyses of the GPCR–arrestin complex[2,17–19]. The chimeric receptor with the V2R C-terminus also allowed us to use the conformation-selective antibody fragment Fab30, which was employed in previous studies to elucidate the activated conformation structures of βarr1[9,11,20]. Since arrestins reportedly interact with membrane lipids when bound to GPCRs[9,10,21], purified β2V2R was embedded within a lipid bilayer of reconstituted high-density lipoproteins (rHDLs)[22]. Considering the facts that acidic lipids reportedly play critical roles in GRK-mediated phosphorylation and arrestin recruitment, and most physiological membranes have a negative net charge, we used a lipid mixture containing acidic lipids to construct rHDLs[23–25]. The reconstituted β2V2R in rHDLs was phosphorylated by GRK2 in a ligand-dependent manner (Supplementary Fig. 1). To prepare phosphorylated β2V2R in rHDLs bound to the inverse agonist, we phosphorylated β2V2R in rHDLs bound to the full agonist, formoterol, by GRK2, and then added excess amounts of the inverse agonist, carazolol, and incubated the reaction until ligand exchange was completed (Fig. 1a, see Methods for details). We confirmed that this procedure results in bound ligand exchange from formoterol to carazolol by monitoring the M82 methyl group resonance of phosphorylated β2V2R, which exhibited distinctly different chemical shifts between the formoterol- and carazolol-bound states (Supplementary Fig. 2).

The interaction between β2V2R in rHDLs and βarr1 was analyzed by surface plasmon resonance (SPR) experiments. The responses of βarr1 to unphosphorylated β2V2R in rHDLs bound to the full agonist were essentially the same as those to empty rHDLs, suggesting that unphosphorylated β2V2R in rHDLs has low affinity for βarr1 (Fig. 1b; middle and right). In contrast, the injection of βarr1 into phosphorylated β2V2R in rHDLs elicited larger responses, suggesting that phosphorylated β2V2R in rHDLs has higher affinity for βarr1 than unphosphorylated β2V2R in rHDLs as expected (Fig. 1b; left and middle). The potentiation of βarr1 affinity upon β2V2R phosphorylation in rHDLs suggests that βarr1 binds to the phosphorylated C tail of β2V2R in rHDLs. Next, to investigate the effect of ligand efficacy on the TM core interaction, we analyzed βarr1 binding to phosphorylated β2V2R in rHDLs bound to full and inverse agonists by SPR experiments. Injections of high concentrations of βarr1 (2.0, 4.0, and 8.0 μM) resulted in serious non-specific binding to the sensor chip surface, thus hampering quantitative analyses of the specific interactions with the receptor (Supplementary Fig. 3). Therefore, the experiments were carried out with a lower concentration range (0.031, 0.063, 0.13, 0.15, 0.5, and 1.0 μM). Steady-state analyses showed that phosphorylated β2V2R in rHDLs bound to the inverse agonist had lower affinity for βarr1 ($K_d > 1$ μM) than that bound to the full agonist ($K_d = 0.24 \pm 0.01$ μM) (Fig. 1C and D). Since the same phosphorylation states of full agonist- and inverse agonist-bound β2V2R in rHDLs were used, the observed affinity variance for βarr1 is strictly due to differences in the ligands bound to the TM core. Furthermore, while the resonances from M215 and M279, located on the intracellular side of the TM core, reportedly exhibited significant chemical shift changes upon βarr1 binding in the full agonist-bound state[26], such changes were not observed upon βarr1 binding in the inverse agonist-bound state (Supplementary Fig. 4), suggesting that βarr1 binding to the TM core is modulated by the ligand efficacy. Therefore, the possibility remains that the C tail undergoes conformational changes in a ligand-dependent manner and subsequently affects

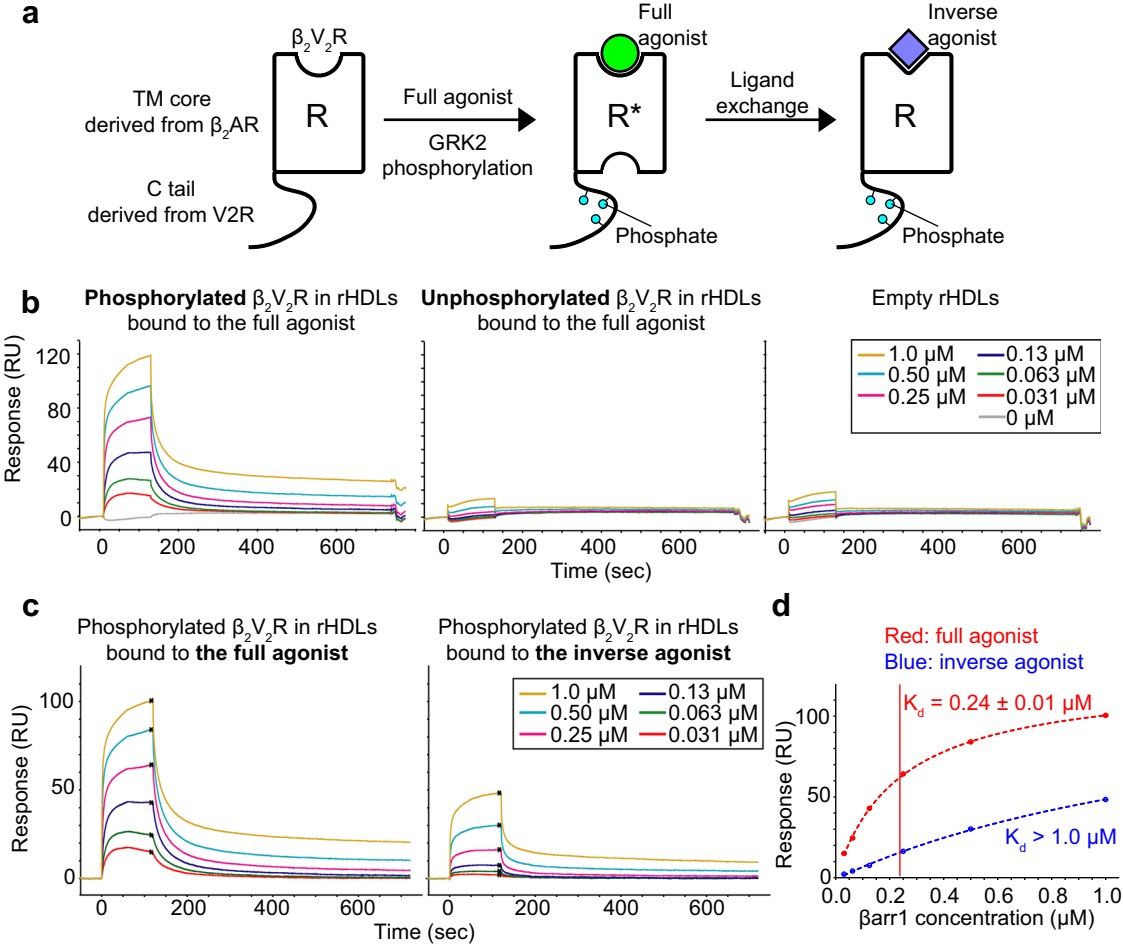

**Fig. 1 SPR analyses of the interactions between βarr1 and β₂V₂R in rHDLs. a** Schematic representation of the various forms of β₂V₂R, composed of the TM core from β₂AR and the C tail from V2R, used in this study. Purified β₂V₂Rs in rHDLs bound to the full agonist are phosphorylated by GRK2. Afterwards, the full agonist was replaced with the inverse agonist. **b** Overlay plots of sensorgrams obtained for the interactions of 0, 0.031, 0.063, 0.13, 0.25, 0.50, and 1.0 μM βarr1 with unphosphorylated β₂V₂R in rHDLs (left), phosphorylated β₂V₂R in rHDLs (middle), and empty rHDLs (right). **c** Overlay plots of sensorgrams obtained for the interactions of 0.031, 0.063, 0.13, 0.25, 0.50, and 1.0 μM βarr1 with phosphorylated β₂V₂R in rHDLs bound to the full agonist (left) and those bound to the inverse agonist (right). To accurately extract the ligand effects, sensorgrams from the reference flow cell immobilized with empty rHDLs were subtracted from those of the active flow cells immobilized with phosphorylated β₂V₂R in rHDLs, and then the sensorgrams from buffer blank injections were subtracted from the reference-subtracted sensorgrams to yield the double-referenced sensorgrams. **d** Plots based on steady-state SPR methods between βarr1 and phosphorylated β₂V₂R in rHDLs bound to the full agonist (red) and the inverse agonist (blue). Each point is the average of 50 data points around the steady state in the sensorgram, and the error bars are their standard deviations. The $K_d$ values were determined by fitting the steady-state response curves, assuming a 1:1 interaction mode.

the interaction with βarr1, it is more likely that βarr1 binds to the TM core of β₂V₂R in rHDLs in a ligand efficacy-dependent manner. Accordingly, βarr1 and β₂V₂R in rHDLs retain the biphasic interaction mode.

**Conformational change of βarr1 induced by both TM core and C tail interactions.** To gain insight into β₂V₂R-induced activation of βarr1, we observed NMR spectra of βarr1 in complex with β₂V₂R in rHDLs. Since the βarr1–β₂V₂R complex embedded in rHDLs has an apparent molecular mass of over 200 kDa, we adopted methyl-selective labeling strategies and applied methyl-TROSY techniques[27]. We prepared isoleucine (Ile) δ1-¹³C¹H-labeled and otherwise perdeuterated βarr1[28], and analyzed their ¹H-¹³C heteronuclear multiple-quantum coherence (HMQC) spectra. βarr1 possesses fifteen Ile residues widely distributed throughout its whole structure (Fig. 2a). Resonance assignments of the fifteen Ile δ1 methyl groups were accomplished by comparing the ¹H-¹³C HMQC spectra of fifteen individual Ile mutants with that of the parental βarr1 construct (Supplementary Fig. 5).

First, we analyzed the conformation of βarr1 without receptors. In the absence of receptors, βarr1 adopts the basal state, in which the inactive conformation is maintained and no signal transduction activity is detected[6]. In the basal state, well-resolved resonances derived from the Ile δ1 methyl groups of βarr1 were observed in the ¹H-¹³C HMQC spectrum (Fig. 2b; left). The resonances from I241 and I317, located on the interface between the N- and C-lobes, were split into two resonances, suggesting that this region exchanges between at least two conformations on a slower timescale than that of the chemical shift difference (Fig. 2b; left). The δ1 methyl group of I377, within the C-terminal region, had an exceptionally narrower resonance than the other methyl groups, indicating the high flexibility of this region (Fig. 2b; left). This observation is in agreement with the fact that the C-terminal region including I377 is disordered in the crystal structure of βarr1 in the basal state (Fig. 2a)[6].

In the ¹H-¹³C HMQC spectrum of βarr1 in the complex with phosphorylated β₂V₂R in rHDLs bound to the full agonist, the overall intensity of the resonances was lower than that in the

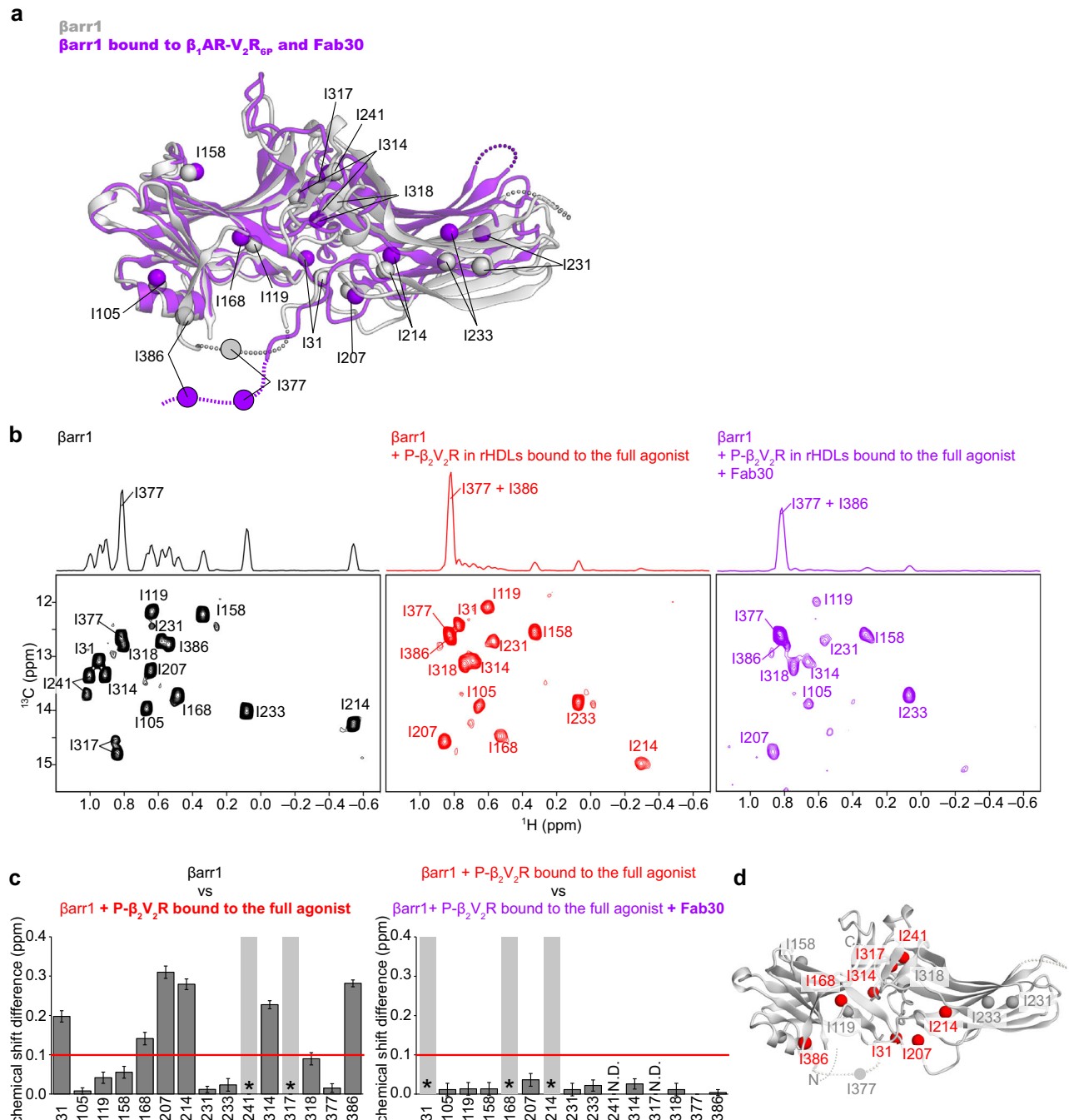

**Fig. 2 Conformational changes of βarr1 upon binding to phosphorylated β₂V₂R in rHDLs bound to the full agonist. a** Structural differences of βarr1 between the basal state (gray, PDB ID: 1G4M) and in complex with β₁AR-V₂R₆P and Fab30 (purple, PDB ID: 6TKO). The structural model was prepared with Cuemol (http://www.cuemol.org/). **b** $^1$H-$^{13}$C HMQC spectra of [u-$^2$H, Ileδ1-$^{13}$C$^1$H$_3$] βarr1 in the basal state (left, black), the complex with phosphorylated β₂V₂R in rHDLs bound to the full agonist (middle, red), and the complex with both phosphorylated β₂V₂R in rHDLs bound to the full agonist and Fab30 (right, purple). **c** Normalized chemical shift differences between the basal state and the complex with phosphorylated β₂V₂R in rHDLs bound to the full agonist (left), and those between the complex with phosphorylated β₂V₂R in rHDLs bound to the full agonist and the complex with both phosphorylated β₂V₂R in rHDLs bound to the full agonist and Fab30 (right). Asterisks indicate residues with resonances broadened beyond detection upon the addition of phosphorylated β₂V₂R in rHDLs bound to the full agonist (left) and those upon the addition of Fab30 (right). N.D. indicates residues with resonances that were not observed both before (B; middle) and after (B; right) Fab30 addition. The error bars were calculated based on the digital resolution of the spectra, as described in "Methods". **d** Distribution of the isoleucine residues on the structure of βarr1 in the basal state (PDB ID: 1G4M). The residues with resonances exhibiting chemical shifts larger than 0.1 ppm upon binding to phosphorylated β₂V₂R in rHDLs bound to the full agonist are shown as red spheres.

spectrum of βarr1 in the basal state, due to an increase in the apparent molecular mass upon complex formation (Fig. 2b; middle). To determine whether this conformational change was induced by lipids alone, the interactions between βarr1 and empty

rHDLs, lacking phosphorylated β₂V₂R, were investigated. No spectral change was detected upon the addition of excess amounts of empty rHDLs to βarr1 (Supplementary Fig. 6). These results suggest that the conformational change of βarr1 is not induced by

interactions with lipids alone, but by those with phosphorylated $\beta_2V_2R$ in rHDLs. To identify which part of βarr1 underwent conformational changes upon complex formation, we calculated the chemical shift differences for the Ile δ1 methyl groups resonances between the basal state and in the complex with phosphorylated $\beta_2V_2R$ in rHDLs bound to the full agonist. Significant chemical shift differences larger than 0.1 ppm were observed for I31, I168, I207, I214, I314, and I386, which are distributed around the interface between the N- and C-lobes (Fig. 2c; left and D). In addition, the resonances from I241 and I317, located on the interface between the N- and C-lobes, were broadened beyond detection in the complex with phosphorylated $\beta_2V_2R$ in rHDLs bound to the full agonist (Fig. 2b; middle). These spectral changes suggested that a conformational change occurred on the interface between the N- and C-lobes of βarr1 upon binding to phosphorylated $\beta_2V_2R$ in rHDLs bound to the full agonist (Fig. 2d). Furthermore, the δ1 methyl groups of I377 and I386, within the C-terminal region, had exceptionally narrower resonances than other methyl groups in the complex with phosphorylated $\beta_2V_2R$ in rHDLs bound to the full agonist, indicating that this region is highly flexible (Fig. 2b; middle and Supplementary Fig. 7). Particularly, the resonance from I386 in the complex with phosphorylated $\beta_2V_2R$ in rHDLs bound to the full agonist was narrower than that in the basal state, indicating that the region including I386 becomes highly flexible upon this interaction (Fig. 2b; middle). This agrees well with the fact that the C-terminal region of βarr1 is released from the N-domain of βarr1 upon interactions with phosphorylated GPCRs (Fig. 2a)[20,29].

To compare the conformation of βarr1 in the complex with phosphorylated $\beta_2V_2R$ in rHDLs bound to the full agonist, observed in our NMR experiments, to the activated conformations previously determined by crystallography and cryo-EM, we used the synthetic antibody fragment, Fab30[9,11,20]. Fab30 specifically recognizes and stabilizes the activated conformation of βarr1, and thus is useful to obtain the NMR spectrum corresponding to the activated conformation[20]. The addition of Fab30 to βarr1, in the complex with phosphorylated $\beta_2V_2R$ in rHDLs bound to the full agonist, induced further intensity reductions and small chemical shift changes in some resonances, suggesting that Fab30 binds to βarr1 in this complex (Fig. 2b; right). However, the observed chemical shift changes upon Fab30 binding were less than 0.04 ppm, and much smaller than those observed upon complex formation with phosphorylated $\beta_2V_2R$ in rHDLs bound to the full agonist (Fig. 2c; right). These results suggest that the conformation of βarr1, in the complex with phosphorylated $\beta_2V_2R$ in rHDLs bound to the full agonist and in the absence of Fab30, is similar to the activated conformation previously determined in the complex with Fab30[9,11,20].

**Partial activation of βarr1 induced by phosphorylated $\beta_2V_2R$ bound to the inverse agonist**. To gain insight into the contributions of the TM core and C tail interactions to the conformational change for βarr1 activation, we obtained the methyl-TROSY spectrum of βarr1 in the complex with phosphorylated $\beta_2V_2R$ in rHDLs bound to the inverse agonist (Fig. 3a). Under these conditions, while the phosphorylation states of the C tail remained the same, the TM core interaction was attenuated relative to the full agonist-bound state, as revealed by the SPR experiments described above (Fig. 1c; right). The result showed that the I158 resonance, in the complex with phosphorylated $\beta_2V_2R$ in rHDLs bound to the inverse agonist, exhibited a chemical shift corresponding to that in the activated conformation (Fig. 3b; upper row). Furthermore, the resonances from I241 and

I317, in the complex with phosphorylated $\beta_2V_2R$ in rHDLs bound to the inverse agonist, were broadened beyond detection, in a similar manner to those in the activated conformation (Fig. 3b; middle and lower rows). These observations suggested that the local environments near the phosphate-binding and TM core-binding regions adopt activated conformations (Fig. 3c). In contrast, the resonances from I119, I168, I214, I231, I314, and I318 exhibited chemical shifts corresponding to the basal conformation, while those from I31, I105, I207, I233, and I386 split into two resonances, one corresponding to the basal conformation and the other to the activated conformation (Fig. 3a). These results suggested that the attenuated core interaction induced by the inverse agonist results in partial activation. In other words, the whole βarr1 structure, except for the phosphate- and TM core-binding regions, exists in equilibrium between the major basal and minor activated conformations.

**Effects of Fab30 on βarr1 in the complex with phosphorylated $\beta_2V_2R$ bound to the inverse agonist**. Our NMR analyses of βarr1 in complex with $\beta_2V_2R$ in rHDLs bound to full- and inverse agonists suggested that while the βarr1 structure adopts the activated conformation in the presence of both the TM core and C tail interactions, the attenuation of the TM core interaction destabilizes the activated conformation. In contrast, the crystal structure of βarr1 in complex with a phosphopeptide derived from V2R (V2Rpp) and Fab30, in which the TM core interactions do not exist, revealed the activated conformation[20]. To determine the reason for this difference, we used methyl-TROSY NMR to investigate whether the conformational change could occur, in βarr1 in the complex with phosphorylated $\beta_2V_2R$ in rHDLs bound to the inverse agonist, upon Fab30 addition. Interestingly, all resonances corresponding to the basal state disappeared upon the addition of Fab30 (Fig. 4a). Furthermore, some resonances, such as I105 and I233, which split into two resonances, completely shifted to those corresponding to the activated conformation (Fig. 4b). These results suggested that, although the TM core interaction is attenuated by the inverse agonist, Fab30 binding promotes βarr1 to adopt the activated-like conformation, in which some parts are similar to those of the activated conformation. This is in agreement with the crystal structure of the βarr1–V2Rpp–Fab30 complex.

## Discussion

GPCR–arrestin complexes are flexible and thus several stabilization methods, such as the use of a GPCR–arrestin fusion protein[8,12], a pre-activated form of arrestin[8,10–12], or a conformation-selective antibody fragment, Fab30[9,11,12], have been utilized to obtain the high resolution structures. Although these methods have provided excellent snapshots of the finally formed complexes, the process of complex formation and conformational activation of arrestins in solution has remained elusive. In this present study, we investigated the interaction between βarr1 and $\beta_2V_2R$ by SPR, and the conformational changes for βarr1 activation without any pre-activated mutation upon binding to $\beta_2V_2R$ embedded in a lipid bilayer, which significantly affects the complex formation[9,11,21], by NMR in a near-physiological solution environment. While βarr1 exhibited little or no affinity for unphosphorylated $\beta_2V_2R$ in rHDLs bound to the full agonist, it exhibited higher affinity for phosphorylated $\beta_2V_2R$ in rHDLs bound to the inverse agonist, as determined by SPR experiments (Fig. 1b; middle, and C; right). These results suggest that the affinity of βarr1 for the phosphorylated C tail is much higher than that for the TM core bound to the full agonist. Our NMR analyses revealed that the inverse agonist-induced attenuation of the TM core interaction destabilized the

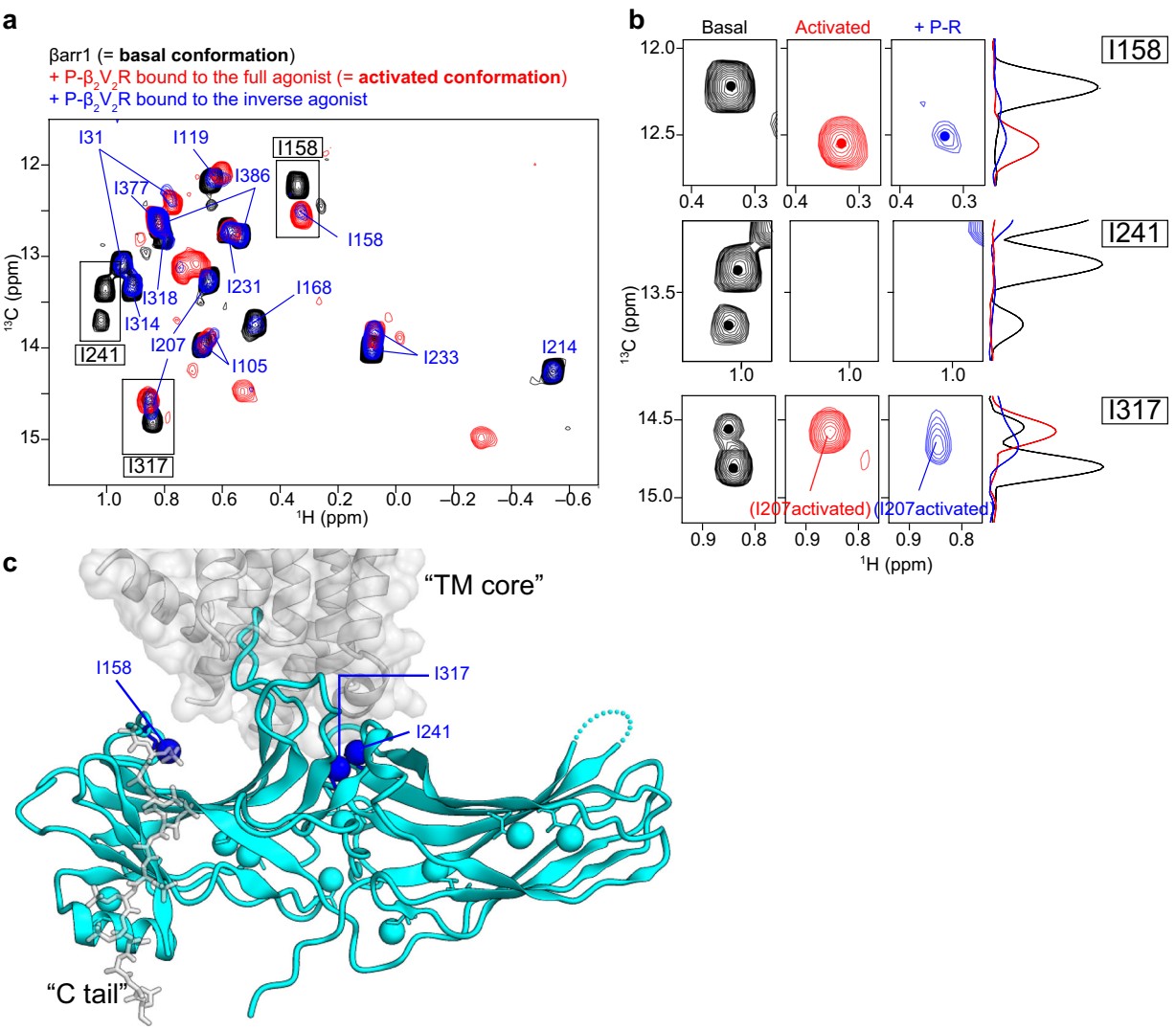

**Fig. 3 Conformational change of βarr1 upon binding to phosphorylated β₂V₂R bound to the inverse agonist. a** Overlay of the $^1$H-$^{13}$C HMQC spectra of [u-$^2$H, Ileδ1-$^{13}$C$^1$H$_3$] βarr1 in the basal state (black), the complex with phosphorylated β₂V₂R in rHDLs bound to the full agonist (red), and the complex with phosphorylated β₂V₂R in rHDLs bound to the inverse agonist (blue). **b** Magnified views of resonances from I158 (upper row), I241 (middle row), and I317 (lower row) in the basal conformation (black), the activated conformation (red), and in the complex with phosphorylated β₂V₂R bound to the inverse agonist (blue). The $^{13}$C 1D projections of selected regions are shown on the right of the spectra. The resonance from I158 exhibited a chemical shift change, and the resonances from I241 and I317 disappeared upon the addition of phosphorylated β₂V₂R in rHDLs bound to the inverse agonist. The split of the resonances from I241 and I317 in the basal state indicated the local conformational exchange, which is distinct from the exchange between the basal and activated conformations. **c** Mapping of the methyl groups adopting the activated conformation, in the complex of phosphorylated β₂V₂R bound to the inverse agonist, on the structure of βarr1 in complex with β₁AR-V2R$_{6P}$ and Fab30 (PDB ID: 6TKO). Methyl groups exhibiting chemical shift changes or signal disappearance upon binding to phosphorylated β₂V₂R in rHDLs bound to the inverse agonist are indicated.

activated conformation, even though the C tail phosphorylation states are retained (Figs. 3 and 5). These results suggest that while the C tail interaction determines the affinity of the complex formation, the TM core interaction plays a crucial role to regulate the conformational changes for βarr1 activation. This agrees with the fact that some GPCRs, which are poorly phosphorylated by GRKs, transiently bind to βarrs but sufficiently activate the signal transduction by the TM core interaction[13,30]. Based on these results, we propose the following biphasic activation model: βarrs first bind to the phosphorylated C tail, which induces local conformational changes around the TM core-binding region, thus enabling the subsequent TM core interaction that stabilizes the activated conformation.

Even though the TM core interaction is attenuated by the inverse agonist, the binding of the conformation-selective antibody fragment, Fab30, can promote βarr1 to adopt the activated-

like conformation, which is similar to the activated conformation (Figs. 4b and 5). This is in good agreement with the fact that βarr1 adopts an activated conformation in the crystal structure of the βarr1–V2Rpp–Fab30 complex, in which the TM core interactions do not exist[20]. Since the addition of Fab30 in the absence of phosphorylated β₂V₂R did not induce significant chemical shift perturbations of the βarr1 resonances, prior interaction with the receptor would be required for Fab30-induced stabilization of the activated-like conformation (Supplementary Fig. 8). The Fab30-induced conformational change toward the activated-like conformation in the absence of the TM core interaction provides a good rationale for the allosteric potentiation of the interaction between βarr1–V2Rpp and ERK2 upon Fab30 binding[31]. Since the extent of the TM core-mediated interaction with βarrs varies among different GPCRs[32], Fab30 may modulate the βarr-mediated signal transduction through GPCRs that poorly

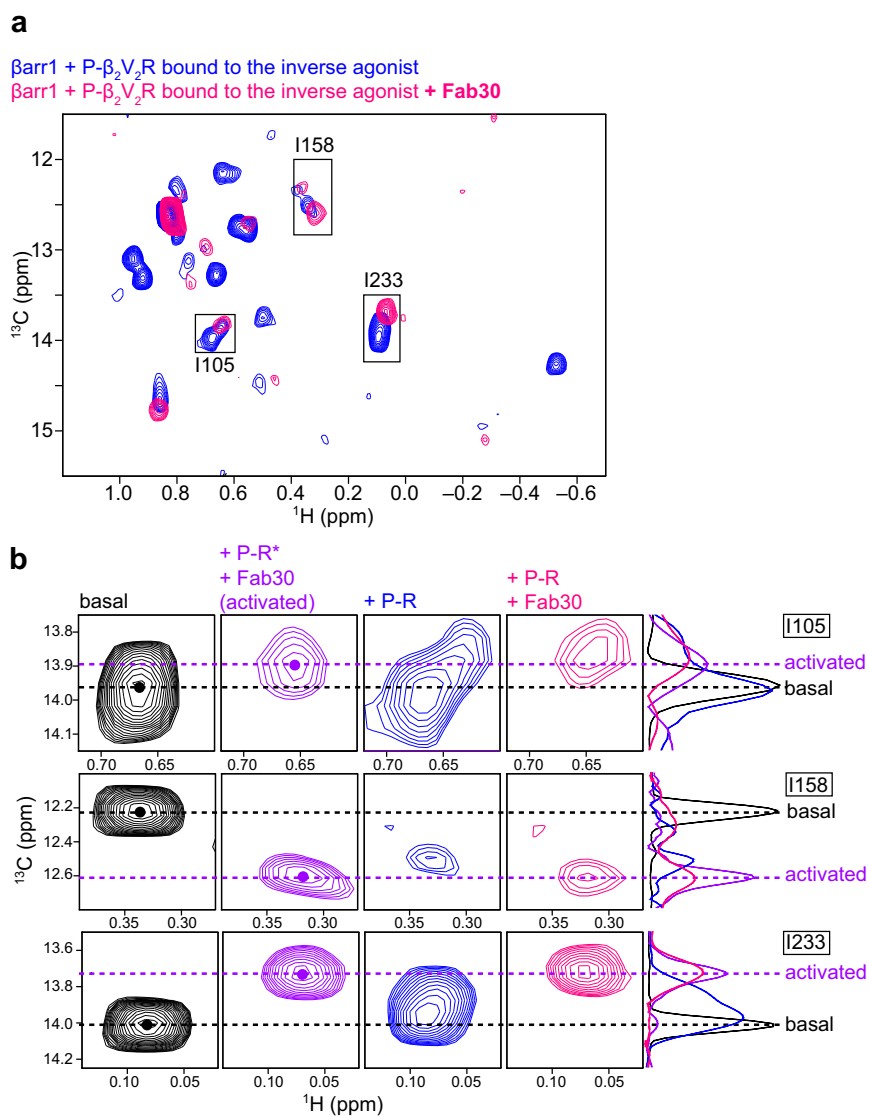

**Fig. 4 Effects of Fab30 on the βarr1 conformation in complex with phosphorylated β₂V₂R bound to the inverse agonist. a** Overlay of the $^{1}$H-$^{13}$C HMQC spectra of [u-$^{2}$H, Ileδ1-$^{13}$C$^{1}$H$_{3}$] βarr1 in the complex with phosphorylated β₂V₂R in rHDLs bound to the inverse agonist (blue) and in the complex with both phosphorylated β₂V₂R in rHDLs bound to the inverse agonist and Fab30 (pink). **b** Comparison of the resonances from the Ileδ1 methyl groups of βarr1 in the basal state (black), in the complex with both phosphorylated β₂V₂R in rHDLs bound to the full agonist and Fab30 (+P-R* + Fab30, purple), in the complex with phosphorylated β₂V₂R in rHDLs bound to the inverse agonist (+P-R, blue), and in the complex with both phosphorylated β₂V₂R in rHDLs bound to the inverse agonist and Fab30 (+P-R + Fab30, pink). The $^{13}$C 1D projections of selected regions are shown on the right of the spectra. The 1D projection of the spectra corresponding to the basal state is reduced by a factor of five, to compare the $^{13}$C chemical shifts clearly. The $^{13}$C chemical shifts corresponding to the basal and activated conformations are indicated by dashed lines.

interact with βarr through the TM core, suggesting potential therapeutic applications.

Our results are closely related to the two different binding modes of the GPCR–βarr complex: fully-engaged state, in which the receptor interacts with βarr through both the C tail and TM core, and partially-engaged state, in which the receptor only interacts with βarr through the C tail[2,19]. The present methyl-TROSY analyses demonstrated that the βarr1 binding mode, in complex with phosphorylated β₂V₂R in rHDLs bound to the inverse agonist, corresponds to the partially-engaged state, since βarr1 in the complex did not affect the conformation of the TM region of phosphorylated β₂V₂R in the inverse agonist-bound state (Supplementary Fig. 4). According to the methyl-TROSY analyses, βarr1 in the partially-engaged state exhibited limited activation, and exists in equilibrium between the major basal and minor activated conformations (Figs. 3a and 5). Considering the

fact that the partially-engaged state is reportedly competent with respect to several βarr-mediated responses, such as receptor internalization and c-Src activation[2,4,5], a small population of the activated conformation is sufficient to recruit the signaling partners related to these responses. The stabilization of the activated conformation in the fully-engaged state observed in the present study may be required to recruit other signaling proteins with lower affinities for βarr.

The ability of synthetic GPCR ligands to induce signal transduction preferentially through either G protein or arrestin, a phenomenon known as biased agonism or functional selectivity, affects the therapeutic properties of ligands and is thus important for drug development[33]. Therefore, characterizations of both the G protein- and arrestin-mediated signaling activities are required to develop drugs targeting GPCRs. In most cases, βarr-mediated signaling activities are evaluated by cell-based βarr-recruitment

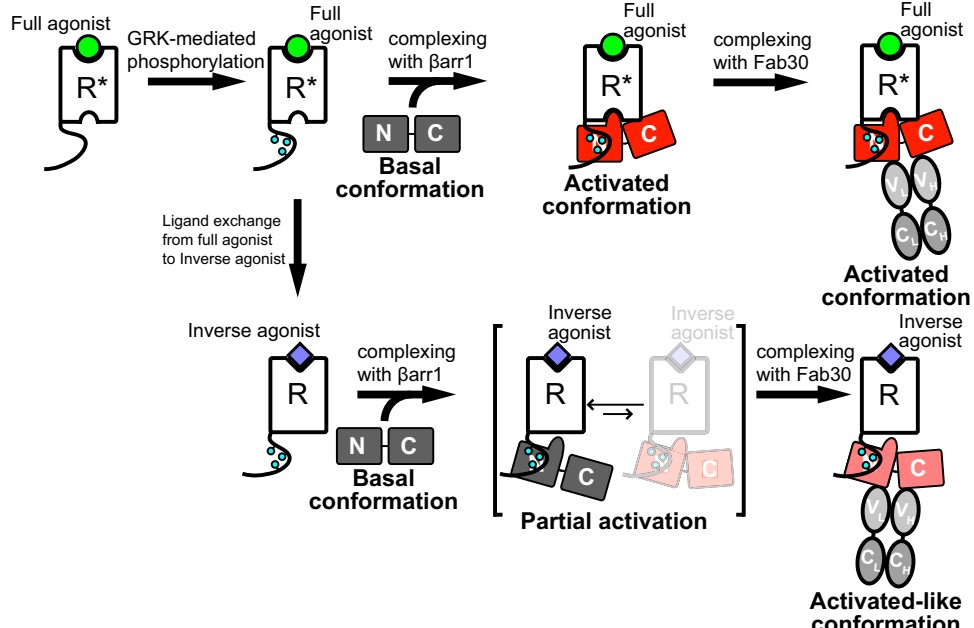

**Fig. 5 Conformational activation of βarr1 upon binding to GPCRs and the synthetic antibody fragment Fab30.** The full agonist-bound receptor is phosphorylated by GRK. The phosphorylated and full agonist-bound receptor interacts with βarr1 through both the TM core and C tail, and βarr1 in the complex adopts the activated conformation. Fab30 binding to the complex does not induce significant conformational changes in βarr1. In the case where the full agonist is replaced with the inverse agonist after GRK-mediated phosphorylation, the receptor only interacts with βarr1 through the C tail, resulting in partial activation in which βarr1 exists in equilibrium between the basal and activated conformations. Fab30 binding to the complex promotes βarr1 to adopt the activated-like conformation, which is partly similar to the activated conformation.

assays, which primarily reflect the ligand-induced GRK-mediated phosphorylation rather than the TM core interaction. Our NMR analyses suggest that the TM core interaction is critical to evaluate the conformational activation of βarr, which would be involved in engaging scaffolding partners. Considering the fact that GPCR ligands control the conformational equilibrium of the transmembrane region[34,35], thus affecting the TM core interaction with βarr, this interaction and subsequent conformational change of βarr should also be evaluated to fully characterize the βarr-mediated signal transduction properties of ligand of interest.

Zhuang *et al.* previously reported a solution NMR investigation of the interactions between visual arrestin (varr), a different subtype from βarr1, and its cognate GPCR rhodopsin in various phosphorylated and activated states[15]. The dark phosphorylated rhodopsin, which was assumed to interact with varr only though the C tail, induced chemical shift changes of the amide resonances from the C-terminal region of varr and the disappearance of the amide resonances from residues around the TM core-binding region. Moreover, Mayer *et al.* reported that phosphopeptides corresponding to the C tail of rhodopsin induced chemical shift changes of the amide resonances from the C-terminal and finger loop regions of varr, and some were split into two resonances, with one corresponding to the basal state[36]. These observations are in line with our results obtained with βarr1 in the complex with phosphorylated $\beta_2V_2R$ bound to the inverse agonist (Fig. 3), suggesting that the effects of C tail interactions on the conformation could be conserved between varr and βarr1. However, light-activated phosphorylated rhodopsin, which was assumed to interact with varr through both the TM core and C tail, induced the disappearance of most of the amide and methyl resonances from varr, except for those from the C-terminal region, suggesting that varr bound to light-activated phosphorylated rhodopsin adopted a dynamic molten globule-like structure[15]. This contrasts with our observation that the methyl resonances from

βarr1, in the complex with phosphorylated $\beta_2V_2R$ bound to the full agonist, could be detected at chemical shifts corresponding to those stabilized by Fab30 (Fig. 2b and c), suggesting that the conformation of βarr1 induced by both the TM core and C tail would be more rigid than that of varr.

Previous conformational analyses of βarrs by solution NMR were based on the site-specific incorporations of unnatural amino acids, such as 3,5-difluorotyrosine (F2Y) and 4-trimethylsilylphenylalanine (TMSiPhe), through genetic code expansion[37,38]. The resonances from these unnatural amino acid sidechains minimally overlap with those derived from proteins and thus can be analyzed in 1D NMR experiments, enabling the rapid detection of conformational change around the labeled site. Using the TMSiPhe probe incorporated into position H295, near the polar core of βarr1, Liu et al. investigated the effect of the TM core of phosphorylated $\beta_2V_2R$ on the conformation around the polar core. They reported that the volume of the resonances corresponding to the active state correlated well with the potency of the ligand, suggesting that the TM core interactions could induce conformational changes around the polar core[38]. Although our Ile probe was located at a different position, the resonance from Ile168, which is also close to the polar core, exhibited a similar ligand-dependent shift from the basal to activated state (Fig. 3). Thus, the present and previous observations are consistent. Our methyl-TROSY analyses also revealed that the TM core-mediated conformational change of βarr1 occurs not only around the polar core, but throughout the entire βarr1 structure.

Pharmacological analyses demonstrated that the allosteric effects of the agonist affinity, induced by βarr1 coupling, vary among different GPCRs[32]. Furthermore, several structures of GPCR–βarr1 complexes revealed the different orientations of βarr1 with respect to the transmembrane region of GPCRs, although the common βarr1 regions are involved[9–12]. Thus, the degree of TM core-induced conformational activation observed in the present study may also vary among different GPCRs. In

addition, conformational analyses based on the Fab30 reactivity suggested that different conformational changes may occur in βarr1 versus βarr2 upon the TM core interaction[39]. Further structural analyses will clarify the details of TM core-induced conformational activation in the GPCR–βarr signaling axis.

## Methods

**Reagents and buffers.** All reagents were from Nacalai Tesque, Inc., or Wako Chemicals unless otherwise noted.

**Generation of recombinant baculovirus expressing β₂V₂R.** The complementary DNA fragment encoding human β₂AR (Met1-C341), with the previously reported mutation (E122W/N187E/C265A)[34], an N-terminal FLAG-tag (DYKDDDDA) and a C-terminal V2R C tail (A343-S371) was cloned into the pFastBac1 vector (Invitrogen). The hemagglutinin signal sequence (MKTIIALSYIFCLVFA) was inserted immediately 5′ to the FLAG tag. Sf9 cells (Invitrogens) were routinely maintained at 27 °C in Grace's supplemented insect cell medium (GIBCO) containing 10% fetal bovine serum (Biowest), 0.1% Pluronic F-68 (GIBCO), 50 International Units mL⁻¹ penicillin, 50 µg mL⁻¹ streptomycin, and 0.125 µg mL⁻¹ amphotericin B. The recombinant baculovirus was generated and amplified with the Bac-to-Bac system (Invitrogen), according to the manufacturer's instructions.

**Expression and purification of β₂V₂R.** The *expres*SF + cells (SF + cells, Protein Science Corp.) were routinely maintained in 100 mL Sf-900 II serum-free medium (GIBCO) in a 250 mL Erlenmeyer flask (Corning), at 27 °C on an orbital shaker (130 rpm). For β₂V₂R expression, the culture was expanded to 3 L. When the cell density reached about $2 \times 10^6$ cells mL⁻¹, 120 mL high-titer virus stocks, 1 µM alprenolol (Sigma-Aldrich), and 14 µM E-64 (Peptide Institute, Inc.) were added, and the culture was continued. Cells were harvested 48 h post-infection by centrifugation at $800 \times g$, and the resulting cell pellets were flash-frozen with liquid nitrogen and stored at −80 °C until use.

All of the following procedures were performed either on ice or in the cold room (4 °C). The cell pellet from 3 L of cell culture was suspended in 300 mL of buffer containing 10 mM HEPES-NaOH (pH 7.2), 20 mM KCl, 10 mM MgCl₂, 0.1 mM tris (2-carboxyethyl) phosphine (TCEP), 1 mM 4-(2-aminoethyl) benzenesulfonyl fluoride hydrochloride (AEBSF), 20 µM leupeptin hemisulfate (Peptide Institute, Inc.), 28 µM E-64 (Peptide Institute, Inc.), and 0.3 µM aprotinin (Wako Chemicals). The cells were disrupted by nitrogen cavitation (Parr Bomb) under 600 psi for 30 min, and the cell lysate was centrifuged at $142,000 \times g$ for 45 min. The resulting pellet was washed twice with 80 mL buffer containing 10 mM HEPES-NaOH (pH 7.2), 1 M NaCl, 20 mM KCl, 10 mM MgCl₂, 0.1 mM TCEP, 1 mM AEBSF, 20 µM leupeptin hemisulfate, and 28 µM E-64, and centrifuged at $142,000 \times g$ for 45 min. The membrane pellet was resuspended in HBSG buffer (20 mM HEPES-NaOH (pH 7.2), 150 mM NaCl, 20% (w/v) glycerol, 0.1 mM TCEP), flash-frozen with liquid nitrogen, and stored at −80 °C.

The membrane pellet from 3 L of cell culture was solubilized for 3 h in 80 mL HBSG buffer supplemented with 1% *n*-dodecyl-β-D-maltopyranoside (DDM, Dojindo), and was then centrifuged at $142,000 \times g$ for 60 min. The supernatant was supplemented with 10 mM CaCl₂, and batch incubated with 8 mL ANTI-FLAG M1 Affinity Agarose Gel (Sigma-Aldrich). The resin was washed with 100 mL HBSG buffer supplemented with 0.1% DDM and 3 mM CaCl₂. The protein was eluted with 20 mL HBSG buffer supplemented with 0.1% DDM, 5 mM EDTA, and 0.1 mg mL⁻¹ DYKDDDDK peptide (Wako Chemicals). The eluted β₂V₂R was concentrated to ~0.5 mL with Amicon Ultra-15 filters (50 kDa molecular weight cut-off, Millipore) and further purified by size exclusion chromatography on a Superose 6 10/300 GL column (GE Healthcare), equilibrated in buffer containing 20 mM HEPES-NaOH (pH 7.2), 150 mM NaCl, 0.1 mM TCEP, and 20 µM formoterol (Toronto Research Chemicals). The eluted β₂V₂R was concentrated to ~0.5 mL with Amicon Ultra-4 filters (30 kDa molecular weight cut-off, Millipore), flash-frozen with liquid nitrogen, and stored at −80 °C until use.

**Preparation of GRK2.** The complementary DNA fragment encoding human GRK2 with a C-terminal hexahistidine-tag was cloned into the pFastBac1 vector (Invitrogen). The recombinant baculovirus was generated and amplified with the Bac-to-Bac system (Invitrogen), according to the manufacturer's instructions. The SF + cells were collected and resuspended in Sf-900 II serum-free medium (GIBCO) at a final cell density of $2 \times 10^6$ cells mL⁻¹. A 12 mL portion of the high-titer virus stock was added to 300 mL of the SF + cell culture, and the culture was continued at 27 °C. The cells were harvested 48 h post-infection by centrifugation at $800 \times g$, and the resulting cell pellets were flash-frozen with liquid nitrogen and stored at −80 °C until use.

All of the following procedures were performed either on ice or in the cold room (4 °C). The cell pellet was resuspended in 100 mL of buffer, containing 20 mM HEPES-NaOH (pH 7.2), 300 mM NaCl, and 0.02% Triton-X100. The cells were disrupted by sonication, and the cell lysate was centrifuged at $100,000 \times g$ for 1 h. The supernatant was mixed with 4 mL of TALON metal affinity resin (Clontech) and batch incubated at 4 °C for 2 h. The resin was washed with 100 mL of buffer, containing 20 mM HEPES-NaOH (pH 7.2), 300 mM NaCl, and 10 mM imidazole. The protein was eluted with 20 mL HEPES-NAOH (pH 7.2), 300 mM NaCl, and 200 mM imidazole. The eluate was concentrated with Amicon Ultra-15 filters (50 kDa molecular weight cut-off, Millipore), and further purified by size exclusion chromatography on a Superdex 200 10/300 GL Increase column (GE Healthcare), equilibrated in buffer containing 20 mM HEPES-NaOH (pH 7.2), 300 mM NaCl, 2 mM EDTA, and 1 mM DTT. The eluted GRK2 was flash-frozen with liquid nitrogen, and stored at −80 °C until use.

**Reconstitution of β₂V₂R into rHDLs.** The *Escherichia coli* BL21 (DE3) (Stratagene), transformed with the plasmid encoding His-tagged MSP1D1 or MSP1E3D1, was cultured at 37 °C in Terrific Bloth (TB) media containing 100 mg L⁻¹ of ampicillin. When the culture attained an OD₆₀₀ of 2.0, 1 mM of IPTG was added to induce protein expression and the culture was continued further at 37 °C for 3 h. The cells were harvested by centrifugation at $5000 \times g$ for 15 min, and the resulting cell pellets were stored at −80 °C. The cells were suspended in buffer containing 50 mM Tris-HCl (pH7.5), 300 mM NaCl, 100 mM KCl, and disrupted by sonication. The cell lysate was centrifuged at $20,000 \times g$ for 30 min, and the pellet was suspended in buffer containing 50 mM Tris-HCl (pH7.5), 300 mM NaCl, 100 mM KCl, 1% Triton X-100. The suspension was centrifuged at $100,000 \times g$ for 1 h, and the supernatant was applied to HIS-Select resin (SIGMA). The resin was washed with buffer containing 50 mM Tris-HCl (pH7.5), 300 mM NaCl, 100 mM KCl, 50 mM Na-cholate, 20 mM imidazole. The His-tagged MSP1D1 or MSP1E3D1 was eluted with buffer containing 50 mM Tris-HCl (pH7.5), 300 mM NaCl, 100 mM KCl, 200 mM imidazole. Afterward, TEV protease was added to the eluate, which was dialyzed against buffer, containing 50 mM Tris-HCl, 10 mM NaCl, 0.5 mM EDTA for 1 day. The sample was then dialyzed against buffer, containing 20 mM HEPES-NaOH (pH 7.2), 150 mM NaCl, and passed through the HIS-Select resin to obtain the MSP1D1 or MSP1E3D1 without His-tag. Since better NMR spectra were obtained when using MSP1D1, we used it to construct rHDLs, except for the NMR experiment in Supplementary Fig. 9 (see Supplementary Note 1 for details).

A mixture of 1-palmitoyl-2-oleoyl-phosphatidylcholine (POPC, Avanti Polar Lipids) and 1-palmitoyl-2-oleoyl-phosphatidylglycerol (POPG, Avanti Polar Lipids) in chloroform was prepared at a molar ratio of 3:2. The solvent was evaporated under a nitrogen atmosphere and dried *in vacuo* to form a lipid film. The film was solubilized in buffer, containing 20 mM HEPES-NaOH (pH 7.2), 150 mM NaCl, 0.1 mM TCEP, and 100 mM sodium cholate, for a final lipid concentration of 50 mM.

All of the following procedures were performed either on ice or in the cold room (4 °C) unless otherwise stated. The MSP1D1 and the lipid solution were added to the prepared β₂V₂R in DDM micelles at final concentrations of 0.1 mM and 5 mM, respectively. The mixture was incubated on ice for at least 3 h. Afterwards, 80% (w/v) of Bio-Beads SM-2 (Bio-Rad) was added to the mixture and incubated at 4 °C overnight with gentle mixing. The supernatant was supplemented with 10 mM CaCl₂, and applied to a 2 mL ANTI-FLAG M1 Affinity Agarose Gel column, equilibrated in HBSG buffer supplemented with 3 mM CaCl₂. The resin was washed with 40 mL HBSG buffer supplemented with 3 mM CaCl₂. The β₂V₂R in rHDLs was eluted with HBSG buffer supplemented with 5 mM EDTA, 0.2 mg mL⁻¹ DYKDDDDK peptide, and 20 µM formoterol, and concentrated to ~0.5 mL with Amicon Ultra-4 filters (10 kDa molecular weight cut-off, Millipore).

For the preparation of phosphorylated β₂V₂R, the buffer was exchanged to 20 mM Tris-HCl, pH 7.5, 10 mM MgCl₂, and 0.1 mM TCEP, using a NAP-5 desalting column (GE Healthcare). Purified GRK2, ATP, and formoterol were added at final concentrations of 2 µM, 1 mM, and 100 µM, respectively. The reaction mixture was incubated at 30 °C for 2 h, and then analyzed by SDS-PAGE with Pro-Q Diamond (Molecular Probes) staining. Gel images were obtained with a Typhoon FLA 9000 imager (GE Healthcare).

The eluate from the ANTI-FLAG M1 Affinity Agarose Gel or the phosphorylation reaction mixture was further purified by size exclusion chromatography on a Superdex 200 10/300 GL Increase column (GE Healthcare), equilibrated in 20 mM HEPES-NaOH (pH 7.2), 150 mM NaCl, 0.1 mM TCEP, and 10 µM formoterol. The eluate was concentrated with Amicon Ultra-4 filters (10 kDa molecular weight cut-off, Millipore), and the buffer was exchanged to 20 mM HEPES-NaOH (pH 7.0), 150 mM NaCl, 0.1 mM TCEP, and 50 µM formoterol, in 100% D₂O.

To prepare phosphorylated β₂V₂R bound to carazolol, the unbound formoterol was removed by using a NAP-5 desalting column (GE Healthcare), and then 100 µM carazolol was added. The reaction was incubated at 298 K for at least 16 h.

**Preparation of isotope-labeled β-arrestin.** The complementary DNA fragment encoding the hexahistidine-tag, the TEV protease-cleavage site (ENLYFQG), and the human β-arrestin 1 cysteine-less mutant[16] was cloned into the pTrcHisB vector (Thermo Fisher Scientific). The resulting protein sequence was: MGGSHHHHHHGMASENLYFQ/GMGDKGTRVFKKASPNGKLTVYLGKRDFV DHIDLVDPVDGVVLVDPEYLKERRVYVTLTVAFRYGREDLDVLGLTFRKDLF VANVQSFPPAPEDKKPLTRLQERLIKKLGEHAYPFTFEIPPNLPSSVTLQPGPED TGKALGVDYEVKAFVAENLEEKIHKRNSVRLVIRKVQYAPERPGPQPTAETT RQFLMSDKPLHLEASLDKEIYYHGEPISVNVHVTNNTNKTVKKIKISVRQYAD IVLFNTAQYKVPVAMEEADDTVAPSSTFSKVYTLTPFLANNREKRGLALDGK LKHEDTNLASSTLLREGANREILGIIVSYKVKVKLVVSRGGLLGDLASSDVAVE

LPFTLMHPKPKEEPPHREVPENETPVDTNLIELDTNDDDIVFEDFARQRLKG MKDDKEEEEDGTGSPQLNNR, where the TEV protease cleaves between Q and G in the cleavage site. Further mutations were introduced by PCR-based site-directed mutagenesis to generate βarr1 variants for the resonance assignments of the iso-leucine δ1 methyl group. The sequences of the oligonucleotides used to generate the βarr1 variants are listed in Supplementary Table 1. The *Escherichia coli* (*E. coli*) BL21 (DE3) Codon Plus RP strain (Stratagene), transformed with the prepared plasmid was cultured overnight at 37 °C in 10 mL Luria Bertani (LB) medium containing 100 mg L$^{-1}$ ampicillin. The overnight culture was used to inoculate 1 L of D$_2$O-based M9 medium, supplemented with 2 g L$^{-1}$ [$^2$H$_7$] glucose (Cambridge Isotope Laboratories) and 1 g L$^{-1}$ $^{15}$NH$_4$Cl (SI Science Co., Ltd.). When the culture attained an OD$_{600}$ of 0.8, 0.3 mM isopropyl-β-D-thiogalactopyranoside (IPTG) was added to induce protein expression and the culture was further incubated at 30 °C for 20 h. For selective $^{13}$C$^1$H$_3$ labeling of the isoleucine δ1 methyl group, the medium was supplemented with 50 mg L$^{-1}$ of [methyl-$^{13}$C, 3,3-$^2$H] ketobutyric acid (Cambridge Isotope Laboratories), 1 h before the IPTG addition[28]. The cells were harvested by centrifugation at 5000 × *g* for 15 min, and the resulting cell pellets were flash-frozen with liquid nitrogen and stored at −80 °C until use.

The cells from a 1 L culture were suspended in 60 mL of 20 mM Tris-HCl (pH 8.0) and 200 mM NaCl, supplemented with Protease Inhibitor Cocktail. The cells were disrupted by sonication, and the cell lysate was centrifuged at 100,000 × *g* for 1 h. The supernatant was applied to a 1 mL HisTrap HP column (GE Healthcare), equilibrated in buffer containing 20 mM Tris-HCl (pH 8.0) and 200 mM NaCl. The protein was eluted by a linear concentration gradient of imidazole, from 25 mM to 175 mM. Afterwards, 1 mg of TEV protease was added to the eluate, which was dialyzed for 16~18 h against buffer containing 20 mM Tris-HCl (pH 8.0), 150 mM NaCl, and 2 mM EDTA. The samples were applied to a 1 mL HiTrap Heparin HP column (GE Healthcare), equilibrated with buffer containing 20 mM Tris-HCl (pH 8.0), 150 mM NaCl, and 2 mM EDTA. The protein was eluted by a linear concentration gradient of NaCl, from 200 mM to 700 mM. The eluate was concentrated with Amicon Ultra-4 filters (30 kDa molecular weight cut-off, Millipore), and further purified by size exclusion chromatography on a Superdex 75 10/300 GL column (GE Healthcare), equilibrated in buffer containing 20 mM Tris-HCl (pH 8.0), 300 mM NaCl, and 2 mM EDTA. The eluate was concentrated with Amicon Ultra-4 filters (10 kDa molecular weight cut-off, Millipore), and the buffer was exchanged to 20 mM HEPES-NaOH (pH 7.0), 150 mM NaCl, 0.1 mM TCEP, and 50 μM β$_2$AR ligand (formoterol or carazolol), 100% D$_2$O.

**Surface plasmon resonance experiments**. The βarr1 binding activities of unphosphorylated and phosphorylated β$_2$V$_2$R were analyzed with a Biacore T200 instrument (GE Healthcare). The β$_2$V$_2$Rs embedded in biotinylated rHDLs were captured on flow cells with the Series S Sensor Chip SA (GE Healthcare), via the interaction between biotin and streptavidin. The binding assays were carried out in buffer containing 20 mM HEPES-NaOH (pH 7.2), 150 mM NaCl, and 1 μM for-moterol or carazolol, at a flow rate of 30 μL min$^{-1}$. The sensorgrams were processed with the Biacore T-200 Evaluation software (GE Healthcare).

**Preparation of Fab30**. *E. coli* 55244 cells (ATCC) were transformed with the Fab30 plasmid, and single colonies were inoculated in 50 mL of 2×YT media and grown overnight at 30 °C as the primary culture. On the next day, secondary cultures were inoculated in fresh 2×YT media with 5% volume of primary culture and grown for 8 h at 30 °C. The cells were then centrifuged, resuspended in an equal volume of CRAP medium, and grown for 16 h at 30 °C. Cell pellets were harvested and stored at −80 °C until further use. For purification, cells were resuspended in lysis buffer (50 mM HEPES-Na$^+$ (pH 8.0), 0.5 M NaCl, 0.5% (v/v) Triton X-100, 0.5 mM MgCl$_2$) and lysed by sonication. The cell lysate was heated at 65 °C in a water bath for 30 min, chilled on ice for 5 min, and then centrifuged at 20,000 × *g* for 30 min. The clarified cell lysate was loaded onto a pre-equilibrated Protein L column at room temperature, which was eluted by gravity flow at a rate of 1~2 mL min$^{-1}$. The Protein L resin was extensively washed with wash buffer (50 mM HEPES-Na$^+$ (pH 8.0), 0.5 M NaCl), and the Fab30 protein was eluted with 100 mM acetic acid. Eluted fractions were neutralized immediately with 10% volume of neutralization buffer (1 M HEPES (pH 8.0)), and subsequently desalted and buffer-exchanged into storage buffer (20 mM HEPES-Na$^+$ (pH 8.0), 0.1 M NaCl) using PD-10 columns (GE Healthcare). Purified Fab30 was supplemented with 10% (v/v) glycerol, flash-frozen, and stored at −80 °C until further use. To assess the functionality of the purified Fab30, a co-immunoprecipitation assay was performed, in which the reactivity of Fab30 towards V2Rpp-bound βarr1 was used as readout. For this assay, purified Fab30 (2.5 μg) was incubated with purified βarr1 (2.5 μg) with or without V2Rpp (pre-incubation with βarr1 for 30 min on ice) in a 100~200 μl reaction volume for 1 h at room temperature, and then pre-equilibrated Protein L beads were added. After an additional 1 h incubation at room temperature, the Protein L beads were washed 3–5 times with wash buffer (20 mM HEPES-Na$^+$ (pH 7.4), 150 mM NaCl, 0.01% (w/v) LMNG), and bound proteins were eluted using 2×SDS loading buffer. Eluted proteins were separated by SDS–PAGE and selective interactions of Fab30 with activated βarr1 were visualized by western blotting and Coomassie Brilliant Blue staining.

**NMR experiments**. All spectra were recorded with a Bruker AVANCE III 800 spectrometer equipped with a cryogenic probe, processed by Topspin 3.6.2 (Bruker), and analyzed by Sparky[40]. The $^1$H chemical shifts were referenced to the methyl protons of 3-(trimethylsilyl)-1-propanesulfonic acid sodium salt, and the $^{13}$C chemical shifts were referenced indirectly.

$^1$H-$^{13}$C heteronuclear multiple-quantum coherence (HMQC) spectra with echo/anti-echo gradient coherence selection were recorded for 15 μM βarr1, in buffer containing 20 mM HEPES-NaOH (pH 7.2), 150 mM NaCl, and 50 μM β$_2$AR ligands prepared in D$_2$O. The phosphorylated β$_2$V$_2$R and Fab30 were mixed at final concentrations of 30 μM and 40 μM, respectively. The spectral widths were set to 16 and 16 ppm for the $^1$H and $^{13}$C dimensions, respectively, and inter-scan delays were set to 1 sec. In total, 1,024 ($^1$H) × 96 ($^{13}$C) complex points were recorded, and 256 scans/FID gave rise to an acquisition time of 14 h for each spectrum. Prior to Fourier transformation, the data matrices were zero-filled to 2,048 ($^1$H) × 128 ($^{13}$C) complex points, and multiplied by a Gaussian apodization function in the $^1$H dimension and a 60°-shifted squared sine bell apodization function in the $^{13}$C dimension.

Normalized chemical shift differences of the Ileδ1 methyl group, $\Delta\delta$, were calculated by the following equation: $\Delta\delta = [(\Delta\delta_{1H})^2 + (\Delta\delta_{13C}/5.7)^2]^{0.5}$. The normalization factor (5.7) is the ratio of the standard deviation of the isoleucine δ1 methyl $^1$H and $^{13}$C chemical shifts, deposited in the Biological Magnetic Resonance Data Bank (http://www.bmrb.wisc.edu/). The error values were calculated by the formula $[\Delta\delta_{1H} R_{1H} + \Delta\delta_{13C} R_{13C}/(5.7)^2]/\Delta\delta$, where $R_{1H}$ and $R_{13C}$ are the digital resolution in ppm in the $^1$H and $^{13}$C dimensions, respectively.

**Reporting summary**. Further information on research design is available in the Nature Research Reporting Summary linked to this article.

## Data availability

The data that support this study are available from the corresponding author upon reasonable request. The chemical shift data have been deposited in Biological Magnetic Resonance Bank under accession code 51131. The previously existing PDB entries 1G4M, 2RH1, and 6TKO were used in this study. Source data are provided with this paper.

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

## Acknowledgements

This work was supported by grants from Japan Society for the Promotion of Science (JSPS) KAKENHI grant number JP17H06097 (to I.S.), JP20H03375 (to T.U.), JP19H04951 (to T.U.), JP21H02410 (to Y.K.), JP20K21473 (to Y.K.), and JP19H04946 (to Y.K.); and the Japan Agency for Medical Research and Development (AMED) grant number JP18ae010104 (to I.S.). Research program in the laboratory of A.K.S. is supported by the Senior Fellowship of DBT/WT India Alliance (IA/S/20/1/504916). A.K.S. is EMBO Young Investigator and Joy Gill Chair Professor. H.D-A. is supported by the BioCare scheme of the Department of Biotechnology (DBT: BT/PR31791/BIC/101/1228/2).

## Author contributions

Y.S., Y.K., T.U., A.K.S., and I.S. designed the research. Y.S. prepared βarr1 and various forms of β₂V₂R embedded in rHDLs, and performed SPR and NMR experiments. S.P. and H.D-A. expressed, purified and characterized Fab30 under the supervision of A.K.S. Y.S., A.K.S., and I.S. analyzed the data and wrote the manuscript.

## Competing interests

The authors declare no competing interests.
