## [Peer Review File · Nature Communications]

Biphasic activation of β -arrestin 1 upon interaction with a GPCR revealed by methyl-TROSY NMRREVIEWER COMMENTS

Reviewer #1 (Remarks to the Author):

Review for NRDD for Shimada paper

Shiraishi et al. present a study of the interactions of arrestin-1 with a chimeric GPCR, β 2AR-V2R, observed by SPR and NMR spectroscopy in aqueous solutions. SPR experiments were done to monitor the response of β arr1 to empty nanodiscs, unphosphorylated β 2V2R, and phosphorylated β 2V2R in complexes with agonists or inverse agonists. HMQC spectra were presented of selectively ^{13}C -labeled arrestin-1 in complexes with the phosphorylated chimeric receptor and an agonist, with the receptor, agonist, and antibody, and in the basal state. A final schematic figure is shown that summarizes the findings from the paper and suggests a mechanism for β arr1 recognition and binding to the receptor that proceeds via a two-step, or biphasic, binding mode involving both elements of the phosphorylated C-terminus and receptor core to achieve "complete" activation.

Overall, the NMR data are of high quality for this challenging system. The data analysis depended on the assignment of 15 individual Ile lines in the NMR spectra, requiring 15 different β arr1 variants to be produced. This step alone is a lot of work and the level of work I would expect from the submitting laboratory. The production of the complex with the phosphorylated receptor and antibody was also clearly a lot of work. The findings support a two-step, or biphasic, mechanism for β arr1 recognition and binding to β 2V2R, which appears consistent with mechanisms proposed from other biophysical and NMR studies (e.g. Reference 33). One of the more novel findings from this manuscript is the experiment using phosphorylated and inverse agonist-bound β 2V2R with the antibody and β arr1. The NMR data appear to suggest that the conformation of β arr1 is highly similar between agonist-bound and inverse agonist-bound phosphorylated β 2V2R, i.e. that Fab30 "forces" the β arr1 to adopt an active conformation. This observation broadens the scope of the presented work and will be of interest to structural biologists and biophysicists who may be thinking about employing similar approaches. The presented data are an interesting and insightful addition to the small but growing body of NMR data on arrestins. Overall I therefore support publication of the presented work once the following comments and questions have been addressed.

Do the authors observe any interactions between β arr1 and Fab30 in absence of the receptor by NMR?

The Discussion section should be expanded to include a closer comparison between the data in the present manuscript with the literature data from References 15 and 33. In Reference 15, the authors of that paper noted a number of chemical shift changes and line broadening observed for β arr1 in complex with phosphorylated rhodopsin. Some of the changes observed in Reference 15 appear consistent with the presented data in the current manuscript, however there do appear to be some differences. The current data should be more carefully compared with the data from Reference 15 and this comparison should be discussed. Regarding Reference 33, the authors from that work studied the same chimeric receptor, arrestin-1, ligands, and even same antibody. Reference 33 used a different NMR approach; nevertheless, the major findings from that paper should also be compared with the main observations from the present text.

Figure 1, panel A, should be modified slightly to be consistent with the chimeric nature of the employed β 2V2R, for example by presenting the C-terminus and receptor core in different colors or labeling them in the figure.

The authors label arrestin-1 in complex with Fab30 and the inverse agonist as an active complex; however, I don't think the current data entirely support this, given that many of the resonances are broadened beyond detection in the spectra for the complexes with Fab30. It may be that arrestin-1 does adopt a similar conformation but there may also be differences that exist that will be revealed in future studies. I suggest the authors relabel the arrestin-1 complex with Fab30 and inverse agonist as "active-like" or something similar to potentially distinguish it from the complex with β 2V2R, agonist

and Fab30.

I think the presentation of the schematic shown in Figure 5 will be somewhat confusing to the average reader. I understand that the authors' intention with this figure is to illustrate the proposed biphasic activation mechanism of arrestin-1. However, the presentation in Figure 5 is neither consistent with the experimental workflow used in this paper nor consistent with our current physiological understanding of this mechanism. The confusing aspect is showing the reaction proceeding from a phosphorylated and inverse agonist-bound receptor to the agonist-bound "activated conformation". The authors should revise this figure to improve the clarity while keeping it consistent with their main findings. One potential way this could be done would be to show the phosphorylation step with agonist-bound receptor and then showing arrows from this state to (1) a different state with the inverse agonist-bound and phosphorylated receptor and (2) the "activated conformation". As a minor comment regarding the presentation of this figure, I don't think the authors can definitively conclude that the conformation of arrestin-1 in complex with agonist-bound receptor is identical to the conformation of arrestin-1 in complex with inverse-agonist bound receptor and antibody are the same (see above). Therefore, the authors should not necessarily label both as "activated conformation" and should instead label the latter conformation as "active-like" or similar.

More details for preparing the sample with inverse agonist and phosphorylated receptor for NMR studies should be included to improve reproducibility. Specifically, the authors state that they added "excess amounts of the inverse agonist, carazolol, and incubated the reaction until the ligand exchange was completed (Fig. 1A)." Exactly how much inverse agonist was used with respect to the agonist concentration? For how long was this mixture incubated? How was the unbound agonist removed from the sample? How was this process monitored to produce a sample for NMR experiments? Finally, how were the authors certain that the ligand exchange was complete and what metrics were used to be sure of this?

Regarding the preparation of the nanodiscs, which variant of MSP1 was used for construction of the lipid nanodiscs (e.g. MSP1D1, MSP1E3D1, etc)? This should be included in the Materials and Methods. The authors should also provide a brief statement either in the Materials and Methods or the main text about why this particular variant was used given that there are now several different choices and it is not clear that measurements on different sized nanodiscs produce the same observations.

Related to the above question, why was the particular lipid mixture used to prepare the nanodiscs? The authors should include a brief statement that justifies this mixture and its relevance to physiology.

The authors should include cross-sections through the selected resonances in the following figures: Figure 3, Figure 4, Supplemental Figure 2, Supplemental Figure 4, and Supplemental Fig 4.

The assignments should be deposited into BMRB before the paper is made available online and in print.

In Materials & Methods, please list the full amino acid sequence of the employed β arr1 protein and list the sequences of the oligonucleotides used to generate the β arr1 variants via site-directed mutagenesis. I realize this requires some additional little work, but this kind of information is typical to see in many journals these days.

Details on the NMR data processing should be provided in the Materials and Methods section.

A brief explanation for how GRK was produced and an appropriate reference or two should be provided in the Materials & Methods section.

Reviewer #2 (Remarks to the Author):

Shirashi et al. present a study of β -arrestin, where they monitor its conformational changes upon binding to the C-terminal tail and the intra-cellular cavity of an activated GPCR. They use a chimeric GPCR consisting of β 2AR fused to the C-terminal tail of vasopressin receptor, which is phosphorylated in vitro by GRK2. Using SPR and NMR they very nicely show differential effects of full and inverse agonists on the final conformation of arrestin in complex with the chimeric GPCR.

I genuinely enjoyed reading this paper and I went through it in one go. The results are significant to the field in that the NMR assays allow dissecting the individual contributions from C-tail and TM core binding and their effects on the activation state of β -arrestin.

The experimental work is very well documented, the results are clearly structured and the conclusions drawn fit with the presented data.

This work leaves only few open questions, which should be addressed by the authors:

Why did the authors not use the native C-tail of β 2AR? The authors previously successfully produced β 2AR with its native C-tail phosphorylated (Nature Comm, 2018), and had obtained indications of arrestin binding, when observing signals of β 2AR. This leads to the question, why β 2AR with the native C-tail was not included in this study? Was initially tried, but the effects were too weak? If such experiments were carried out, is it possible to learn something about the native situation?

The description of the SPR measurements is a bit short. What were the exact concentrations used for the individual curves in the experiments shown in Fig. 1C? Without these, one can not judge the quality of the K_d determination.

Were there any significant differences in on- and off-rates? In the inverse agonist case, arrestin seems to have a faster off-rate, which could be used to further narrow down the K_d in this case and to quantify the contribution to the affinity from the TM core.

Mayer et al. (Nature Comm. 2019, 10:1261) studied the interaction of arr-1 with different phosphorylated C-tail peptides by NMR and found a similar gradual activation of arr-1. Interestingly, also here, the largest chemical shift changes were observed in the finger loop, and weaker effects extended to the C-domain. The similarities and differences of arr-1 and β -arrestin 1 should be discussed (e.g. in the paragraph starting at line 272)

The text in the two lower right panels in Figure 3B is too small. Additionally, please add labels to the boxes in Panel 1A. Residue 158 is labeled, while 241 and 317 are not.

In general, the use of articles in the text is sometimes wrong. Please revise the English during the corrections of the manuscript.

Reviewer #3 (Remarks to the Author):

In this manuscript, the authors report the interactions between β -arrestin 1 and a re-constructed GPCR to analyze the conformational changes of β -arrestin 1 during its activation by phosphorylated GPCR, which were monitored by methyl-TROSY based NMR spectroscopy, the state-of-art technique. overall, this is a nice study of β -arrestin 1 by NMR and SPR technique and the results are very interesting. This manuscript provides important new insights into the activation of β -arrestin 1 at the atomic level.

The manuscript is recommended to be published in this journal after addressing the reviewer's concerns.

Questions for the authors:

1. From the SPR experiments, the authors conclude that the β -arrestin 1 binds to the TM core of β 2V2R in a ligand efficacy-dependent manner supported by the SPR results using different ligands. Is there any possibility that the C tail changes its conformation state (though this region is disordered) when binds to different ligands, which subsequently affect the interaction between β -arrestin 1 and the

C tail? In such case, it is not only the TM core, the C tail also has such ligand-dependent manner on binding to β -arrestin 1.

2. In the second part of RESULTS, "Conformational change of β -arrestin 1 induced by both TM core and C tail interactions", the authors claim that β -arrestin 1 does not interact with lipid (line 138-141). Is there any control experiment performed to monitor the interaction between β -arrestin 1 and empty lipid? Is there the possibility that in the present experiment the β -arrestin 1 was saturated already and did not observe the interaction between them when adding extra lipid?

3. For the conformational analysis by NMR, the methyl group of I377 of β -arrestin 1 which locates at disordered C tail, showed a narrower peak when there is no receptors. However, did this peak disappeared after β -arrestin 1 interacts with β 2V2R? Since the authors indicated that the C tail region is released from the N-domain of β -arrestin 1 upon interact with the GPCRs and another residue I386 at this region could be observed. The authors should have an explanation about this observation.

Point-by-point responses to the reviewers' comments

We would like to thank the reviewers for their comments and suggestions for our manuscript. According to the reviewers' comments, we revised the manuscript carefully. Our point-by point responses are summarized below. The Reviewer's comments are shown in *italics*. The page and line numbers represent those in the source DOCX file.

Reviewer #1 (Remarks to the Author):

Review for NRDD for Shimada paper

Shiraishi et al. present a study of the interactions of arrestin-1 with a chimeric GPCR, β 2AR-V2R, observed by SPR and NMR spectroscopy in aqueous solutions. SPR experiments were done to monitor the response of β arr1 to empty nanodiscs, unphosphorylated β 2V2R, and phosphorylated β 2V2R in complexes with agonists or inverse agonists. HMQC spectra were presented of selectively ^{13}C -labeled arrestin-1 in complexes with the phosphorylated chimeric receptor and an agonist, with the receptor, agonist, and antibody, and in the basal state. A final schematic figure is shown that summarizes the findings from the paper and suggests a mechanism for β arr1 recognition and binding to the receptor that proceeds via a two-step, or biphasic, binding mode involving both elements of the phosphorylated C-terminus and receptor core to achieve "complete" activation.

Overall, the NMR data are of high quality for this challenging system. The data analysis depended on the assignment of 15 individual Ile lines in the NMR spectra, requiring 15 different β arr1 variants to be produced. This step alone is a lot of work and the level of work I would expect from the submitting laboratory. The production of the complex with the phosphorylated receptor and antibody was also clearly a lot of work. The findings support a two-step, or biphasic, mechanism for β arr1 recognition and binding to β 2V2R, which appears consistent with mechanisms proposed from other biophysical and NMR studies (e.g. Reference 33). One of the more novel findings from this manuscript is the experiment using phosphorylated and inverse agonist-bound β 2V2R with the antibody and β arr1. The NMR data appear to suggest that the conformation of β arr1 is highly similar

between agonist-bound and inverse agonist-bound phosphorylated β_2V_2R , i.e. that Fab30 “forces” the $\beta arr1$ to adopt an active conformation. This observation broadens the scope of the presented work and will be of interest to structural biologists and biophysicists who may be thinking about employing similar approaches. The presented data are an interesting and insightful addition to the small but growing body of NMR data on arrestins. Overall I therefore support publication of the presented work once the following comments and questions have been addressed.

We appreciate the reviewer for his/her positive comments.

<Comment 1-1>

Do the authors observe any interactions between $\beta arr1$ and Fab30 in absence of the receptor by NMR?

We appreciate the reviewer’s valuable question about the properties of the $\beta arr1$ –Fab30 interaction. According to the reviewer’s question, we investigated the interaction between $\beta arr1$ and Fab30 in the absence of β_2V_2R by an NMR titration experiment, as shown in Supplementary Figure 8. Upon the addition of 1.5, 3.0, 4.5, and 6.5 equivalent amounts of Fab30 to 17 μM $\beta arr1$, significant intensity reductions were observed, suggesting that Fab30 weakly binds to $\beta arr1$ in absence of β_2V_2R (apparent $K_d > 20 \mu M$). However, significant chemical shift changes were not observed for all methyl resonances except for the resonance from I233, which split into two upon the addition of Fab30. These results suggest that Fab30 changes the microenvironment around I233, but does not induce a large conformational change of $\beta arr1$ in the absence of β_2V_2R . In other words, prior full or partial conformational activation induced by the receptor is required for further stabilization by Fab30. This is in good agreement with the previous report that the stable interaction between $\beta arr1$ and Fab30 requires V_2Rpp (Shukla *et al.*, *Nature* (2013), 497, 137-141).

In the revised manuscript, we added a description about the results in the discussion section and Supplementary Fig. 8, as follows.

(Page 15, lines 249-252)

“Since the addition of Fab30 in the absence of phosphorylated β_2V_2R did not induce significant chemical shift perturbations of the $\beta arr1$ resonances, prior interaction with the

receptor would be required for Fab30-induced stabilization of the activated-like conformation (Supplementary Fig. 8).”

Supplementary Figure 8 | βarr1 –Fab30 interactions in the absence of $\beta_2\text{V}_2\text{R}$. Overlay of ^1H - ^{13}C HMQC spectra of [$u\text{-}^2\text{H}$, Ile δ 1- $^{13}\text{C}^1\text{H}_3$] βarr1 in the basal state (black) and in the presence of 3.0 molar equivalents of Fab30 (red). Overlaid projections in the presence of 0.0 (black), 1.5 (cyan), 3.0 (red), 4.5 (violet), and 6.5 (green) molar equivalents of Fab30 are displayed above the spectra.

<Comment 1-2>

The Discussion section should be expanded to include a closer comparison between the data in the present manuscript with the literature data from References 15 and 33. In Reference 15, the authors of that paper noted a number of chemical shift changes and line broadening observed for βarr1 in complex with phosphorylated rhodopsin. Some of the changes observed in Reference 15 appear consistent with the presented data in the current manuscript, however there do appear to be some differences. The current data should be more carefully compared with the data from Reference 15 and this comparison should

be discussed. Regarding Reference 33, the authors from that work studied the same chimeric receptor, arrestin-1, ligands, and even same antibody. Reference 33 used a different NMR approach; nevertheless, the major findings from that paper should also be compared with the main observations from the present text.

We appreciate the reviewer's suggestion. According to the recommendations from Reviewer #1 and Comment 2-4 from Reviewer #2, we discussed the comparison with previous studies, such as Ref. 15, Ref. 36, and Ref. 38 (Ref. 33 in the initial manuscript), as follows.

(Pages 17-18, lines 287-306)

“Zhuang *et al.* previously reported a solution NMR investigation of the interactions between visual arrestin (varr), a different subtype from β arr1, and its cognate GPCR rhodopsin in various phosphorylated and activated states [15]. The dark phosphorylated rhodopsin, which was assumed to interact with varr only through the C tail, induced chemical shift changes of the amide resonances from the C terminal region of varr and the disappearance of the amide resonances from residues around the TM core-binding region. Moreover, Mayer *et al.* reported that phosphopeptides corresponding to the C tail of rhodopsin induced chemical shift changes of the amide resonances from the C terminal and finger loop regions of varr, and some were split into two resonances, with one corresponding to the basal state [36]. These observations are in line with our results obtained with β arr1 in the complex with phosphorylated β_2V_2R bound to the inverse agonist (Fig. 3), suggesting that the effects of C tail interactions on the conformation could be conserved between varr and β arr1. However, light-activated phosphorylated rhodopsin, which was assumed to interact with varr through both the TM core and C tail, induced the disappearance of most of the amide and methyl resonances from varr, except for those from the C terminal region, suggesting that varr bound to light-activated phosphorylated rhodopsin adopted a dynamic molten globule-like structure [15]. This contrasts with our observation that the methyl resonances from β arr1, in the complex with phosphorylated β_2V_2R bound to the full agonist, could be detected at chemical shifts corresponding to those stabilized by Fab30 (Fig. 2B and C), suggesting that the conformation of β arr1 induced by both the TM core and C tail would be more rigid than that of varr.”

(Pages 18-19, lines 312-321)

“Using the TMSiPhe probe incorporated into position H295, near the polar core of β arr1,

Liu *et al.* investigated the effect of the TM core of phosphorylated β_2V_2R on the conformation around the polar core. They reported that the volume of the resonances corresponding to the active state correlated well with the potency of the ligand, suggesting that the TM core interactions could induce conformational changes around the polar core [38]. Although our Ile probe was located at a different position, the resonance from Ile168, which is also close to the polar core, exhibited a similar ligand-dependent shift from the basal to activated state (Fig. 3). Thus, the present and previous observations are consistent. Our methyl-TROSY analyses also revealed that the TM core-mediated conformational change of β arr1 occurs not only around the polar core, but throughout the entire β arr1 structure”

<Comment 1-3>

Figure 1, panel A, should be modified slightly to be consistent with the chimeric nature of the employed β_2V_2R , for example by presenting the C-terminus and receptor core in different colors or labeling them in the figure.

We appreciate the reviewer’s suggestion. According to the reviewer’s suggestion, we modified the Figure 1A and Figure legends to clarify that the receptor we used in this study is chimeric composed of the TM core derived from β_2AR and the C tail derived from V2R, as follows.

Figure 1 | SPR analyses of the interactions between β arr1 and β_2V_2R in rHDLs. (A) Schematic representation of the various forms of β_2V_2R , composed of the TM core from β_2AR and the C tail from V2R, used in this study. Purified β_2V_2R s in rHDLs bound to the full agonist are phosphorylated by GRK2. Afterwards, the full agonist was replaced with the inverse agonist.

<Comment 1-4>

The authors label arrestin-1 in complex with Fab30 and the inverse agonist as an

active complex; however, I don't think the current data entirely support this, given that many of the resonances are broadened beyond detection in the spectra for the complexes with Fab30. It may be that arrestin-1 does adopt a similar conformation but there may also be differences that exist that will be revealed in future studies. I suggest the authors relabel the arrestin-1 complex with Fab30 and inverse agonist as "active-like" or something similar to potentially distinguish it from the complex with B2V2R, agonist and Fab30.

We agree with the reviewer's careful comment that the possibility remains that the conformation of β arr1 in complex with phosphorylated β 2V2R bound to the inverse agonist and Fab30 is not completely identical the activated conformation. According to the reviewer's suggestion, we now refer to it as "activated-like conformation". In the revised manuscript, we modified the sentences describing the activated-like conformation, as follows.

(Page 2, lines 27-29)

"The conformation-selective antibody, Fab30, forces partially activated β arr into the activated-like conformation."

(Page 13, lines 216-218)

"These results suggested that, although the TM core interaction is attenuated by the inverse agonist, Fab30 binding forces β arr1 to adopt the activated-like conformation, in which some parts are similar to those of the activated conformation."

(Page 15, lines 245-247)

"Even though the TM core interaction is attenuated by the inverse agonist, the binding of the conformation-selective antibody fragment, Fab30, can force β arr1 to adopt the activated-like conformation, which is similar to the activated conformation (Figs. 4B and 5)"

(Page 15, lines 252-255)

"The Fab30-induced conformational change toward the activated-like conformation in the absence of the TM core interaction provides a good rationale for the allosteric potentiation of the interaction between β arr1-V2Rpp and ERK2 upon Fab30 binding [31]."

<Comment 1-5>

I think the presentation of the schematic shown in Figure 5 will be somewhat confusing to the average reader. I understand that the authors' intention with this figure is to illustrate the proposed biphasic activation mechanism of arrestin-1. However, the presentation in Figure 5 is neither consistent with the experimental workflow used in this paper nor consistent with our current physiological understanding of this mechanism. The confusing aspect is showing the reaction proceeding from a phosphorylated and inverse agonist-bound receptor to the agonist-bound "activated confirmation". The authors should revise this figure to improve the clarity while keeping it consistent with their main findings. One potential way this could be done would be to show the phosphorylation step with agonist-bound receptor and then showing arrows from this state to (1) a different state with the inverse agonist-bound and phosphorylated receptor and (2) the "activated conformation". As a minor comment regarding the presentation of this figure, I don't think the authors can definitively conclude that the conformation of arrestin-1 in complex with agonist-bound receptor is identical to the conformation of arrest-1 in complex with inverse-agonist bound receptor and antibody are the same (see above). Therefore, the authors should not necessarily label both as "activated conformation" and should instead label the latter confirmation as "active-like" or similar.

We appreciate the reviewer's suggestion that Figure 5 should be revised for average readers to improve its clarity while keeping it consistent with the main findings. According to the reviewer's suggestion, we added the phosphorylation step and the ligand-exchange step to Figure 5. We also re-labeled β arr1 in the complex with phosphorylated β_2V_2R bound to the inverse agonist and Fab30 as the "activated-like conformation", according to Comment 1-4. We believe that the presentation in the revised Figure 5 is now consistent with our experimental workflow.

Figure 5 | Conformational activation of β arr1 upon binding to GPCRs and the synthetic antibody fragment Fab30. The full agonist-bound receptor is phosphorylated by GRK. The phosphorylated and full agonist-bound receptor interacts with β arr1 through both the TM core and C tail, and β arr1 in the complex adopts the activated conformation. Fab30 binding to the complex does not induce significant conformational changes in β arr1. In the case where the full agonist is replaced with the inverse agonist after GRK-mediated phosphorylation, the receptor only interacts with β arr1 through the C tail, resulting in partial activation in which β arr1 exists in equilibrium between the basal and activated conformations. Fab30 binding to the complex forces β arr1 to adopt the activated-like conformation, which is partly similar to the activated conformation.

<Comment 1-6>

More details for preparing the sample with inverse agonist and phosphorylated receptor for NMR studies should be included to improve reproducibility. Specifically, the authors state that they added “excess amounts of the inverse agonist, carazolol, and incubated the reaction until the ligand exchange was completed (Fig. 1A).” Exactly how much inverse agonist was used with respect to the agonist concentration? For how long was this mixture incubated? How was the unbound agonist removed from the sample? How was this process monitored

to produce a sample for NMR experiments? Finally, how were the authors certain that the ligand exchange was complete and what metrics were used to be sure of this?

We agree with the reviewer's suggestion. In the revised manuscript, we included the detailed procedure for preparing the phosphorylated β_2V_2R bound to the inverse agonist in the Methods section, as follows.

(Page 25, lines 435-437)

“To prepare phosphorylated β_2V_2R bound to carazolol, the unbound formoterol was removed by using a NAP-5 desalting column (GE Healthcare), and then 100 μM carazolol was added. The reaction was incubated at 298 K for at least 16 h.”

To investigate whether this procedure results in the exchange of the bound ligand, we monitored the resonances from the M82 methyl group of phosphorylated β_2V_2R , whose chemical shift in the formoterol-bound state ($M82^A$) was remarkably different from that in the carazolol-bound state ($M82^U$ and $M82^D$). In the spectrum measured immediately after the addition of carazolol to the formoterol-bound β_2V_2R , we observed an intense resonance corresponding to $M82^A$, but the resonances corresponding to $M82^U$ or $M82^D$ were not detected, suggesting that most β_2V_2R s at this time point were in the formoterol-bound state (Supplementary Fig. 2, black). On the other hand, in the spectrum measured at 16 h after the addition of 100 μM carazolol, the resonance corresponding to $M82^A$ was no longer detected, and resonances from $M82^U$ and $M82^D$ were observed, suggesting that most β_2V_2R s after a 16 h incubation with carazolol were in the carazolol-bound state (Supplementary Fig. 2, red). Therefore, we concluded that this procedure results in the complete exchange of the bound ligand from formoterol to carazolol.

In the revised manuscript, we added Supplementary Fig. 2 and the following sentence describing that the ligand exchange was confirmed by monitoring the M82 methyl resonance of β_2V_2R .

(Pages 6-7, lines 92-95)

“We confirmed that this procedure results in bound ligand exchange from formoterol to carazolol by monitoring the M82 methyl group resonance of phosphorylated β_2V_2R , which exhibited distinctly different chemical shifts between the formoterol- and

carazolol-bound states (Supplementary Fig. 2).”

Supplementary Figure 2 | Confirmation of ligand exchange as monitored by the resonances from the M82 methyl group of phosphorylated β_2V_2R in rHDLs. Overlay of 1H - ^{13}C HMQC spectra of [2H -9AA, $\alpha\beta\gamma$ - 2H , ϵ - ^{13}C -Met] phosphorylated β_2V_2R bound to formoterol immediately (black) and 16 h (red) after carazolol addition. The acquisition time was 3 h for each spectrum. Previous studies reported that the M82 methyl group resonances exhibited distinctly different chemical shifts between the formoterol-bound ($M82^A$) and carazolol-bound ($M82^U$ and $M82^D$) states [1, 2]. Cross sections of the resonances corresponding to $M82^A$ and $M82^U$ are shown on the right.

<Comment 1-7>

Regarding the preparation of the nanodiscs, which variant of MSP1 was used for construction of the lipid nanodiscs (e.g. MSP1D1, MSP1E3D1, etc)? This should be included in the Materials and Methods. The authors should also provide a brief statement either in the Materials and Methods or the main text about why this particular variant was used given that there are now several different choices and it is not clear that measurements on different sized nanodiscs produce the same observations.

We used MSP1D1 for the construction of the nanodiscs in this study, based on the following reason. Previous studies reported by us and other

groups indicated that GPCRs in nanodiscs assembled with MSP1D1 (e.g., Shiraishi *et al.*, *Nat. Commun.* (2018), 9, 194) and MSP1E3D1 (e.g., Bayburt *et al.*, *J. Biol. Chem.* (2011) 286, 1420-1428) were able to interact with arrestins. The reported Stokes diameters of nanodiscs constructed by MSP1D1 and MSP1E3D1 are 9.7 nm and 12.1 nm, respectively (Denisov *et al.*, *J. Am. Chem. Soc.* (2004) 126, 3477-3487). At the early stage of the project, to determine whether the difference in the nanodisc sizes could affect the β arr1 conformation, we compared the NMR spectrum of β arr1 in the complex with phosphorylated β_2V_2R bound to the full agonist in rHDLs assembled with MSP1D1 with that in rHDLs assembled with MSP1E3D1 (Supplementary Fig. 9). The observed chemical shift differences were small (<0.05 ppm), suggesting that the conformation of β arr1 in the complex with phosphorylated β_2V_2R bound to the full agonist in rHDLs assembled with MSP1D1 was almost identical to that in rHDLs assembled with MSP1E3D1 (Supplementary Fig. 9B). On the other hand, more intense resonances were obtained when using MSP1D1, probably because the size of rHDLs assembled with MSP1D1 are smaller than those with MSP1E3D1 (Supplementary Fig. 9A). Therefore, except for the NMR experiment in Supplementary Fig. 9, MSP1D1 was used to construct rHDLs.

In the revised manuscript, we described the MSP variant used in this study, and the reason why we chose MSP1D1 in the Methods section and Supplementary Notes, as follows.

(Page 23, lines 417-419)

“MSP1D1 and MSP1E3D1 were prepared as described previously [40]. Since better NMR spectra were obtained when using MSP1D1, we used it to construct rHDLs, except for the NMR experiment in Supplementary Fig. 9 (see Supplementary Notes for details).”

(Supplementary Notes)

“Effects of the rHDL sizes on the β arr1 conformation in the complex with phosphorylated β_2V_2R in rHDLs bound to the full agonist.

To determine whether differences in the rHDL sizes could affect the β arr1 conformation, the NMR spectrum of β arr1 in the complex with phosphorylated β_2V_2R bound to the full agonist in rHDLs assembled with MSP1D1 was compared with that in rHDLs assembled with MSP1E3D1 (Supplementary Fig. 9). The observed chemical shift differences were small (<0.05 ppm), suggesting that the β arr1 conformation in the complex with phosphorylated β_2V_2R bound to the full agonist in rHDLs assembled with

MSP1D1 was almost identical to that in rHDLs assembled with MSP1E3D1 (Supplementary Fig. 9B). More intense resonances were obtained when using MSP1D1, probably because the rHDLs assembled with MSP1D1 are smaller than those with MSP1E3D1 (Supplementary Fig. 9A). Therefore, except for the NMR experiment in Supplementary Fig. 9, MSP1D1 was used to construct rHDLs.”

Also, we added a new figure showing the comparison of MSP1D1 and MSP1E3D1, as Supplementary Fig.9.

Supplementary Figure 9 | Effects of the rHDL sizes on the βarr1 conformation in the complex with phosphorylated β₂V₂R in rHDLs bound to the full agonist. (A) Overlay of ¹H-¹³C HMQC spectra of [u-²H, Ileδ1-¹³C¹H₃] βarr1 in the complex with phosphorylated β₂V₂R bound to the full agonist, in rHDLs assembled with MSP1D1 (black) and rHDLs assembled with MAP1E3D1 (red). The ¹H 1D projections are shown above the spectra, and a magnified view of the projection including resonances from I158, I214, and I233 is shown in the inset. (B) Normalized chemical shift differences of βarr1 methyl groups between the complex with phosphorylated β₂V₂R bound to the full agonist in rHDLs assembled with MSP1D1 and the complex with phosphorylated β₂V₂R bound to the full agonist in rHDLs assembled with MSP1E3D1. The error bars were calculated based on the digital resolution of the spectra.

<Comment 1-8>

Related to the above question, why was the particular lipid mixture used to

prepare the nanodiscs? The authors should include a brief statement that justifies this mixture and its relevance to physiology.

We appreciate the reviewer's suggestion. According to the reviewer's suggestion, we added the following sentence to describe the reason why we used the lipid mixture containing acidic lipids.

(Page 6, lines 85-88)

“Considering the facts that acidic lipids reportedly play critical roles in GRK-mediated phosphorylation and arrestin recruitment, and most physiological membranes have a negative net charge, we used a lipid mixture containing acidic lipids to construct rHDLs [23-25].”

<Comment 1-9>

The authors should include cross-sections through the selected resonances in the following figures: Figure 3, Figure 4, Supplemental Figure 2, Supplemental Figure 4, and Supplemental Fig 4.

We agree with the author's suggestion to include the cross-sections of the resonances. According to the reviewer's suggestion, in the revised manuscript, we modified Figure. 3, Figure 4, Supplementary Figure 5 (corresponding to Supplementary Fig. 2 in the initial manuscript), and Supplementary Figure 4, as follows.

Figure 3 | Conformational change of β arr1 upon binding to phosphorylated β_2V_2R bound to the inverse agonist. (A) Overlay of the 1H - ^{13}C HMQC spectra of [u - 2H , Ile δ 1- ^{13}C 1H_3] β arr1 in the basal state (black), the complex with phosphorylated β_2V_2R in rHDLs bound to the full agonist (red), and the complex with phosphorylated β_2V_2R in rHDLs bound to the inverse agonist (blue). (B) Magnified views of resonances from I158 (upper row), I241 (middle row), and I317 (lower row) in the basal conformation (black), the activated conformation (red), and in the complex with phosphorylated β_2V_2R bound to the inverse agonist (blue). The ^{13}C 1D projections of selected regions are shown on the right of the spectra. The resonance from I158 exhibited a chemical shift change, and the resonances from I241 and I317 disappeared upon the addition of phosphorylated β_2V_2R in rHDLs bound to the inverse agonist. The split of the resonances from I241 and I317 in the basal state indicated the local conformational exchange, which is distinct from the exchange between the basal and activated conformations. (C) Mapping of the methyl groups adopting the activated conformation, in the complex of phosphorylated β_2V_2R bound to the inverse agonist, on the structure of

β arr1 in complex with β_1 AR-V2R_{6P} and Fab30 (PDB ID: 6TKO). Methyl groups exhibiting chemical shift changes or signal disappearance upon binding to phosphorylated β_2 V₂R in rHDLs bound to the inverse agonist are indicated.

Figure 4 | Effects of Fab30 on the β arr1 conformation in complex with phosphorylated β_2 V₂R bound to the inverse agonist. (A) Overlay of the ^1H - ^{13}C HMQC spectra of [u - ^2H , Ile δ 1- $^{13}\text{C}^1\text{H}_3$] β arr1 in the complex with phosphorylated β_2 V₂R in rHDLs bound to the inverse agonist (blue) and in the complex with both phosphorylated β_2 V₂R in rHDLs bound to the inverse agonist and Fab30 (pink). (B) Comparison of the resonances from the Ile δ 1 methyl groups of β arr1 in the basal state (black), in the complex with both phosphorylated β_2 V₂R in rHDLs bound to the full

agonist and Fab30 (+P-R*+Fab30, purple), in the complex with phosphorylated β_2V_2R in rHDLs bound to the inverse agonist (+P-R, blue), and in the complex with both phosphorylated β_2V_2R in rHDLs bound to the inverse agonist and Fab30 (+P-R+Fab30, pink). The ^{13}C 1D projections of selected regions are shown on the right of the spectra. The 1D projection of the spectra corresponding to the basal state is reduced by a factor of five, to compare the ^{13}C chemical shifts clearly. The ^{13}C chemical shifts corresponding to the basal and activated conformations are indicated by dashed lines.

Supplementary Figure 5 | Assignment of Ile δ 1 methyl resonances of [u - ^2H , Ile δ 1- $^{13}\text{C}^1\text{H}_3$] β arr1. ^1H - ^{13}C HMQC spectra of β arr1 and its mutants, in which isoleucine residues are substituted with other amino acids, are overlaid in black and red, respectively. Resonances assigned to mutated residues are labeled. Cross sections of assigned

resonances at the dashed lines are shown in the insets.

Supplementary Figure 4 | Effect of β arr1 binding on the conformation of the transmembrane region of phosphorylated β_2V_2R bound to an inverse agonist. (A) Distribution of the NMR probes on the structure of the β_2AR TM region. The crystal structure of β_2AR bound to an inverse agonist, carazolol, (PDB ID: 2RH1) [3], is shown as a ribbon model, and the methionine residues observed in the experiments are depicted by sticks. M215 and M279, with resonances that reportedly exhibit significant chemical shift changes upon β arr1 binding in the full agonist-bound state, are highlighted [4]. (B) Overlay of 1H - ^{13}C HMQC spectra of [2H -9AA, $\alpha\beta\gamma$ - 2H , ϵ - ^{13}C -Met] phosphorylated β_2V_2R bound to the inverse agonist (black) and in the complex with β arr1 (red). (C–E) 1H -1D cross-sections of resonances from M215 and M279. The intensity reductions are probably due to the increased apparent molecular weight upon β arr1 binding.

<Comment 1-10>

The assignments should be deposited into BMRB before the paper is made available online and in print.

We agree with the reviewer's comment. We deposited our assignment of the Ile δ 1 methyl resonances of β arr1 to BMRB, with the entry code 51131. The entry will be kept on hold until the manuscript is available.

<Comment 1-11>

In Materials & Methods, please list the full amino acid sequence of the employed

βarr1 protein and list the sequences of the oligonucleotides used to generate the *βarr1* variants via site-directed mutagenesis. I realize this requires some additional little work, but this kind of information is typical to see in many journals these days.

We agree with the reviewer’s suggestion to list the full amino acid sequence of *βarr1* and the oligonucleotide sequences used to generate the *βarr1* variants. According to the reviewer’s suggestion, we added the amino acid sequence of *βarr1* in the Methods section, as follows.

(Pages 25-26, lines 442-451)

“The resulting protein sequence was:

MGGSHHHHHHGMASEENLYFQ/GMGDKGTRVFKKASPNGKLTVYLGKRDFVDH
 IDLVDPVDGVVLVDPEYLKERRVYVTLTVAFRYGREDLDVLGLTFRKDLFVANV
 QSFPPAPEDKKPLTRLQERLIKKLGEHAYPFTFEIPPNLPSVTLQPGPEDTGKAL
 GVDYEVKAFVAENLEEKIHKRNSVRLVIRKVVQYAPERPGPQPTAETTRQFLMSD
 KPLHLEASLDKEIYYHGEPISVNVHVTNNTNKT VKKIKISVRQYADIVLFNTAQY
 KVPVAMEEADDTVAPSSSTFSKVYTLTPFLANNREKRGLALDGLKHEDTNLASS
 TLLREGANREILGIIVSYKVKVVLVSRGGLLDLASSDVAVELPFTLMHPKPKE
 EPPHREVPENETPVDTNLIELDTNDDDIVFEDFARQLKGMKDDKEEEEDGTGS
 PQLNNR, where the TEV protease cleaves between Q and G in the cleavage site
 (underlined).”

We also added the descriptions of the *βarr1* variants in the Methods section and listed the oligonucleotide sequences used to generate the variants in supplementary Table 1.

(Page 26, lines 451-454)

“Further mutations were introduced by PCR-based site-directed mutagenesis to generate *βarr1* variants for the resonance assignments of the isoleucine δ1 methyl group. The sequences of the oligonucleotides used to generate the *βarr1* variants are listed in Supplementary Table 1.”

Supplementary Table 1 | Oligonucleotide sequences used to generate *βarr1* variants

	DNA sequence (5’-3’)
I31L_forward	CTTTGTGGACCACCTCGACCTCGTGGACCCT

I31L_reverse	AGGGTCCACGAGGTGAGGTGGTCCACAAAG
I105L_forward	GCAGGAACGCCTCTCAAGAAGCTGGGCG
I105L_reverse	CGCCCAGCTTCTTGAGGAGGCGTTCCTGC
I119L_forward	CCCTTTCACCTTTGAGCTCCCTCCAAACCTTCC
I119L_reverse	GGAAGGTTTGGAGGGAGCTCAAAGGTGAAAGGG
I158S_forward	GAATTTGGAGGAGAAGAGCCACAAGCGGAATTCTG
I158S_reverse	CAGAATTCCGCTTGTGGCTCTTCTCCTCCAAATTC
I168V_forward	CTGTGCGTCTGGTCCTCCGGAAGGTTTCAGTATG
I168V_reverse	CATACTGAACCTTCCGGAGGACCAGACGCACAG
I207L_forward	CCTCTCTGGATAAGGAGCTCTATTACCATGGAG
I207L_reverse	CTCCATGGTAATAGAGCTCCTTATCCAGAGAGG
I214L_forward	CCATGGAGAACCCCTCAGCGTCAACGTCC
I214L_reverse	GGACGTTGACGCTGAGGGGTTCTCCATGG
I231L_forward	GGAGACGGTGAAGAAGCTCAAGATCTCAGTGC
I231L_reverse	GCACTGAGATCTTGAGCTTCTTACCCTCTCC
I233V_forward	GGTGAAGAAGATCAAGGTCTCAGTGCGCCAG
I233V_reverse	CTGGCGCACTGAGACCTTGATCTTCTTCACC
I241V_forward	GCGCCAGTATGCAGACGTCGTCCTTTTCAACAC
I241V_reverse	GTGTTGAAAAGGACGACGTCTGCATACTGGCGC
I314V_forward	GCCAACCGTGAGGTCTGGGGATCATTG
I314V_reverse	CAATGATCCCCAGGACCTCACGGTTGGC
I317M_forward	CCGTGAGATCCTGGGGATGATTGTTTCTTAC
I317M_reverse	GTAGGAAACAATCATCCCCAGGATCTCACGG
I318V_forward	GAGATCCTGGGGATCGTTGTTTCTTACAAAGTG
I318V_reverse	CACTTTGTAGGAAACAACGATCCCCAGGATCTC
I377L_forward	CCAGTAGATACCAATCTCCTAGAACTTGACACAAATGATG
I377L_reverse	CATCATTTGTGTCAAGTTCTAGGAGATTGGTATCTACTGG
I386L_forward	GACACAAATGATGACGACCTTGTATTTGAGGACTTTGC
I386L_reverse	GCAAAGTCCTCAAATACAAGGTCGTCATCATTTGTGTC

<Comment 1-12>

Details on the NMR data processing should be provided in the Materials and Methods section.

We thank the reviewer for pointing out that the description of the NMR data processing was missing in our initial manuscript. According to the reviewer's comment, in the revised manuscript, we added the following sentence

about the NMR data processing details in the Methods section, as follows.

(Page 30, lines 532-535)

“Prior to Fourier transformation, the data matrices were zero-filled to 2,048 (^1H) \times 128 (^{13}C) complex points, and multiplied by a Gaussian apodization function in the ^1H dimension and a 60°-shifted squared sine bell apodization function in the ^{13}C dimension.”

<Comment 1-13>

A brief explanation for how GRK was produced and an appropriate reference or two should be provided in the Materials & Methods section.

We thank the reviewer for pointing out that the description of the preparation of GRK was missing in the Methods section. In the revised manuscript, we added the explanation of the procedure for preparing GRK2, as follows.

(Pages 22-23, lines 380-400)

“Preparation of GRK2

The complementary DNA fragment encoding human GRK2 with a C-terminal hexahistidine-tag was cloned into the pFastBac1 vector (Invitrogen). The recombinant baculovirus was generated and amplified with the Bac-to-Bac system (Invitrogen), according to the manufacturer’s instructions. The SF+ cells were collected and resuspended in Sf-900 II serum-free medium (GIBCO) at a final cell density of 2×10^6 cells mL^{-1} . A 12 mL portion of the high-titer virus stock was added to 300 mL of the SF+ cell culture, and the culture was continued at 27 °C. The cells were harvested 48 h post-infection by centrifugation at $800 \times g$, and the resulting cell pellets were flash-frozen with liquid nitrogen and stored at -80 °C until use.

All of the following procedures were performed either on ice or in the cold room (4 °C). The cell pellet was resuspended in 100 mL of buffer, containing 20 mM HEPES-NaOH (pH 7.2), 300 mM NaCl, and 0.02% Triton-X100. The cells were disrupted by sonication, and the cell lysate was centrifuged at $100,000 \times g$ for 1 h. The supernatant was mixed with 4 mL of TALON metal affinity resin (Clontech) and batch incubated at 4 °C for 2 h. The resin was washed with 100 mL of buffer, containing 20 mM HEPES-NaOH (pH 7.2), 300 mM NaCl, and 10 mM imidazole. The protein was eluted with 20 mL HEPES-NAOH (pH 7.2), 300 mM NaCl, and 200 mM imidazole. The eluate was concentrated with Amicon Ultra-15 filters (50 kDa molecular weight cut-off, Millipore),

and further purified by size exclusion chromatography on a Superdex 200 10/300 GL Increase column (GE Healthcare), equilibrated in buffer containing 20 mM HEPES-NaOH (pH 7.2), 300 mM NaCl, 2 mM EDTA, and 1 mM DTT. The eluted GRK2 was flash-frozen with liquid nitrogen, and stored at -80°C until use.”

Reviewer #2 (Remarks to the Author):

Shirashi et al. present a study of β -arrestin, where they monitor its conformational changes upon binding to the C-terminal tail and the intra-cellular cavity of an activated GPCR. They use a chimeric GPCR consisting of β 2AR fused to the C-terminal tail of vasopressin receptor, which is phosphorylated in vitro by GRK2. Using SPR and NMR they very nicely show differential effects of full and inverse agonists on the final conformation of arrestin in complex with the chimeric GPCR. I genuinely enjoyed reading this paper and I went through it in one go. The results are significant to the field in that the NMR assays allow dissecting the individual contributions from C-tail and TM core binding and their effects on the activation state of β -arrestin.

The experimental work is very well documented, the results are clearly structured and the conclusions drawn fit with the presented data.

This work leaves only few open questions, which should be addressed by the authors:

We are grateful to the reviewer for his/her positive comments.

<Comment 2-1>

Why did the authors not use the native C-tail of β 2AR? The authors previously successfully produced β 2AR with its native C-tail phosphorylated (Nature Comm, 2018), and had obtained indications of arrestin binding, when observing signals of β 2AR. This leads to the question, why β 2AR with the native C-tail was not included in this study? Was initially tried, but the effects were too weak? If such experiments were carried out, is it possible to learn something about the native situation?

We appreciate the reviewer’s question about the reason why we used the chimeric receptor instead of the native β 2AR. In this study, we used the chimeric receptor β 2V2R in order to compare our NMR results with structural insights

revealed by electron microscopy, since many were obtained by using a chimeric receptor with V2R C-tail and Fab30 (e.g., Shukla *et al.*, *Nature* (2014) 512, 218-222, Staus *et al.*, *Nature* (2020) 579, 297-302, Lee *et al.*, *Nature* (2020), 583, 862-866). Particularly, since Fab30 is a specific antibody fragment to the V2Rpp-bound β arr1, it is reasonable to use the chimeric receptor with the V2R C tail.

We have not yet tried to investigate the conformational change of β arr1 upon the interaction with native β_2 AR. However, as the reviewer pointed out, such experiments would be valuable to understand the activation of β arr1 in the native situation, and we would like to try it in a future study. Close comparisons of the results obtained from the native and chimeric receptors would provide the insights into the effects of the C tail sequence on the conformational changes of β arr1.

In the revised manuscript, we added the following description to clarify the reason why we used the chimeric receptor.

(Page 6, lines 79-83)

“This receptor construct reportedly has higher affinity for β arr1 than wild-type β_2 AR, and thus was used in previous structural analyses of the GPCR-arrestin complex [2, 17-19]. The chimeric receptor with the V2R C-terminus also allowed us to use the conformation-selective antibody fragment Fab30, which was employed in previous studies to elucidate the activated conformation structures of β arr1 [9, 11, 20].”

<Comment 2-2>

The description of the SPR measurements is a bit short. What were the exact concentrations used for the individual curves in the experiments shown in Fig. 1C? Without these, one can not judge the quality of the K_d determination.

The β arr1 concentrations used in the experiments shown in Fig. 1C were 0.031, 0.063, 0.13, 0.25, 0.50, and 1.0 μ M, and the apparent K_d was determined by steady-state analyses. In the full agonist case, the determined K_d (0.24 ± 0.01 μ M) was sufficiently lower than the maximum concentration used in the experiment, suggesting that the K_d was reliable. On the other hand, in the inverse agonist case, the calculated K_d value was 1.6 ± 0.3 μ M, which was higher than the maximum concentration used in the experiments. The experiments with higher β arr1 concentrations were hampered by the large responses derived

from the non-specific binding to the sensor chip surface, as shown in Supplementary Fig. 3. Thus, while we believe that the K_d value in the inverse agonist case was higher than 1 μM , it is difficult to further narrow down the K_d value in our assay.

Supplementary Figure 3 | Effects of non-specific binding of β arr1 to the sensor chip surface. Overlay plots of sensorgrams obtained upon injections of 0, 1.0, 2.0, 4.0, and 8.0 μM β arr1 into flow cells without immobilized molecules.

In the revised manuscript, we added a description about the exact concentration of β arr1 and the reason why we used this concentration range, as follows.

(Pages 7-8, lines 106-112)

“Injections of high concentrations of β arr1 (2.0, 4.0, and 8.0 μM) resulted in serious non-specific binding to the sensor chip surface, thus hampering quantitative analyses of the specific interactions with the receptor (Supplementary Fig. 3). Therefore, the experiments were carried out with a lower concentration range (0.031, 0.063, 0.13, 0.15, 0.5, and 1.0 μM). Steady-state analyses showed that phosphorylated $\beta_2\text{V}_2\text{R}$ in rHDLs bound to the inverse agonist had lower affinity for β arr1 ($K_d > 1 \mu\text{M}$) than that bound to the full agonist ($K_d = 0.24 \pm 0.01 \mu\text{M}$) (Fig. 1C and D).”

Also, we modified Figure 1B and C to show the exact concentrations of β arr1 in the experiments, and added the steady state plots of the SPR experiments as Figure 1D to clarify the quality of the determined K_d , as follows.

Figure 1 | SPR analyses of the interactions between β arr1 and β_2V_2R in rHDLs. (B) Overlay plots of sensorgrams obtained for the interactions of 0, 0.031, 0.063, 0.13, 0.25, 0.50, and 1.0 μM β arr1 with unphosphorylated β_2V_2R in rHDLs (left), phosphorylated β_2V_2R in rHDLs (middle), and empty rHDLs (right). (C) Overlay plots of sensorgrams obtained for the interactions of 0.031, 0.063, 0.13, 0.25, 0.50, and 1.0 μM β arr1 with phosphorylated β_2V_2R in rHDLs bound to the full agonist (left) and those bound to the inverse agonist (right). To accurately extract the ligand effects, sensorgrams from the reference flow cell immobilized with empty rHDLs were subtracted from those of the active flow cells immobilized with phosphorylated β_2V_2R in rHDLs, and then the sensorgrams from buffer blank injections were subtracted from the reference-subtracted sensorgrams to yield the double-referenced sensorgrams. (D) Plots based on steady-state SPR methods between β arr1 and phosphorylated β_2V_2R in rHDLs bound to the full agonist (red) and the inverse agonist (blue). Each point is the average of 50 data points around the steady state in the sensorgram, and the error bars are their standard deviations. The K_d values were determined by fitting the steady-state response curves, assuming a 1:1 interaction mode.

<comment 2-3>

Were there any significant differences in on- and off-rates? In the inverse agonist case, arrestin seems to have a faster off-rate, which could be used to further narrow down the K_d in this case and to quantify the contribution to the affinity

from the TM core.

We appreciate the reviewer's comment that the on- and off-rates should be analyzed to quantify their contributions to the affinity from the TM core. The dissociation phases of the sensorgrams shown in Fig. 1C indicate that the off-rates in the inverse agonist-bound state seem to be faster than those in the full agonist-bound state. Regrettably, however, we could not quantify the on- and off-rates accurately, since the sensorgrams shown in Fig. 1C poorly fit with a simple 1:1 Langmuir model ($\chi^2 = 20.3$ in the full agonist case). Furthermore, as shown below, the dissociation phase of the sensorgrams, with their vertical axis displayed on a logarithmic scale, did not seem indicate a linear function, suggesting that the dissociation reaction is not described by a first-order reaction.

Dissociation phases of the sensorgrams with the vertical axis displayed on a logarithmic scale. (Left) Full agonist-bound state. (Right) Inverse agonist-bound state. The analyte concentrations are 0.25 μM in both panels.

In such cases, the more complex binding model is generally chosen, based on the biological characteristics of the system (Lipschultz *et al.*, *Methods* (2000) 10-310-318). However, we currently lack biological evidence to determine the plausible binding model. Although the two-state model described in the previous study gave better fits ($\chi^2 = 7.1$ in the full agonist case) than the Langmuir 1:1 model ($\chi^2 = 20.3$ in the full agonist case), this was not sufficient experimental proof for the model selection (Lipschultz *et al.*, *Methods* (2000) 10-310-318).

For these reasons, while we believe that the apparent K_d in the full agonist

case ($0.24 \pm 0.01 \mu\text{M}$) was at least 4 times lower than that in the inverse agonist case ($> 1 \mu\text{M}$), unfortunately we cannot be more quantitative than the current interpretation.

<Comment 2-4>

Mayer et al. (Nature Comm. 2019, 10:1261) studied the interaction of arr-1 with different phosphorylated C-tail peptides by NMR and found a similar gradual activation of arr-1. Interestingly, also here, the largest chemical shift changes were observed in the finger loop, and weaker effects extended to the C-domain. The similarities and differences of arr-1 and β -arrestin 1 should be discussed (e.g. in the paragraph starting at line 272)

We appreciate the reviewer's suggestion that the similarities and differences of arr-1 and β arr1 should be discussed, based on previous studies. According to the suggestion from this reviewer and comment 1-2 from Reviewer # 2, we described the similarities and differences of our results obtained by using the β arr1- $\beta_2\text{V}_2\text{R}$ system and the previous studies obtained by using visual arrestin and C terminal peptides derived from rhodopsin (Ref. 36, Mayer *et al.*, *Nat. Commun.* (2019) 10: 1261) or intact rhodopsin (Ref. 15, Zhuang *et al.*, *Proc. Natl. Acad. Sci. USA* (2013) 110, 942-947), as follows.

(Pages 17-18, lines 287-306)

“Zhuang *et al.* previously reported a solution NMR investigation of the interactions between visual arrestin (varr), a different subtype from β arr1, and its cognate GPCR rhodopsin in various phosphorylated and activated states [15]. The dark phosphorylated rhodopsin, which was assumed to interact with varr only through the C tail, induced chemical shift changes of the amide resonances from the C terminal region of varr and the disappearance of the amide resonances from residues around the TM core-binding region. Moreover, Mayer *et al.* reported that phosphopeptides corresponding to the C tail of rhodopsin induced chemical shift changes of the amide resonances from the C terminal and finger loop regions of varr, and some were split into two resonances, with one corresponding to the basal state [36]. These observations are in line with our results obtained with β arr1 in the complex with phosphorylated $\beta_2\text{V}_2\text{R}$ bound to the inverse agonist (Fig. 3), suggesting that the effects of C tail interactions on the conformation could be conserved between varr and β arr1. However, light-activated phosphorylated rhodopsin, which was assumed to interact with varr through both the TM core and C tail,

induced the disappearance of most of the amide and methyl resonances from *varr*, except for those from the C terminal region, suggesting that *varr* bound to light-activated phosphorylated rhodopsin adopted a dynamic molten globule-like structure [15]. This contrasts with our observation that the methyl resonances from β arr1, in the complex with phosphorylated β_2V_2R bound to the full agonist, could be detected at chemical shifts corresponding to those stabilized by Fab30 (Fig. 2B and C), suggesting that the conformation of β arr1 induced by both the TM core and C tail would be more rigid than that of *varr*.

<Comment 2-5>

The text in the two lower right panels in Figure 3B is too small. Additionally, please add labels to the boxes in Panel 1A. Residue 158 is labeled, while 241 and 317 are not.

According to the reviewer's comment, we enlarged the text of Figure 3B so it is the same size as the other text in the same Figure. We also added labels indicating I241 and I317.

Figure 3 | Conformational change of βarr1 upon binding to phosphorylated β₂V₂R bound to the inverse agonist. (A) Overlay of the ¹H-¹³C HMQC spectra of [u-²H, Ileδ1-¹³C¹H₃] βarr1 in the basal state (black), the complex with phosphorylated β₂V₂R in rHDLs bound to the full agonist (red), and the complex with phosphorylated β₂V₂R in rHDLs bound to the inverse agonist (blue). (B) Magnified views of resonances from I158 (upper row), I241 (middle row), and I317 (lower row) in the basal conformation (black), the activated conformation (red), and in the complex with phosphorylated β₂V₂R bound to the inverse agonist (blue). The ¹³C 1D projections of selected regions are shown on the right of the spectra. The resonance from I158 exhibited a chemical shift change, and the resonances from I241 and I317 disappeared upon the addition of phosphorylated β₂V₂R in rHDLs bound to the inverse agonist. The split of the resonances from I241 and I317 in the basal state indicated the local conformational exchange, which is distinct from the exchange between the basal and activated conformations. (C) Mapping of the methyl groups adopting the activated conformation, in the complex of phosphorylated β₂V₂R bound to the inverse agonist, on the structure of

β arr1 in complex with β_1 AR-V2R_{6P} and Fab30 (PDB ID: 6TKO). Methyl groups exhibiting chemical shift changes or signal disappearance upon binding to phosphorylated β_2 V₂R in rHDLs bound to the inverse agonist are indicated.

<Comment 2-6>

In general, the use of articles in the text is sometimes wrong. Please revise the English during the corrections of the manuscript.

We thank the reviewer for pointing out the usage of the articles in the text. The revised manuscript was received English proofreading before the submission.

Reviewer #3 (Remarks to the Author):

In this manuscript, the authors report the interactions between β -arrestin 1 and a re-constructed GPCR to analyze the conformational changes of β -arrestin 1 during its activation by phosphorylated GPCR, which were monitored by methyl-TROSY based NMR spectroscopy, the state-of-art technique. overall, this is a nice study of β -arrestin 1 by NMR and SPR technique and the results are very interesting. This manuscript provides important new insights into the activation of β -arrestin 1 at the atomic level.

The manuscript is recommended to be published in this journal after addressing the reviewer's concerns.

We appreciate the reviewer for his/her positive comments.

<Comment 3-1>

Questions for the authors:

1. From the SPR experiments, the authors conclude that the β -arrestin 1 binds to the TM core of β_2 V₂R in a ligand efficacy-dependent manner supported by the SPR results using different ligands. Is there any possibility that the C tail changes its conformation state (though this region is disordered) when binds to different ligands, which subsequently affect the interaction between β -arrestin 1 and the C tail? In such case, it is not only the TM core, the C tail also has such ligand-dependent manner on binding to β -arrestin 1.

We thank the reviewer for pointing out the possibility that the C tail could

contribute to the ligand-dependent potentiation of the β arr1- β_2 V₂R interaction. Based on the present data, we cannot completely rule out the possibility that the conformation of the phosphorylated C tail changes upon binding to different ligands.

While we agree with the possibility that the C tail could contribute to the ligand-dependent modulation of the β arr1- β_2 V₂R interaction, we believe that the interaction between β arr1 and the TM core of β_2 V₂R is modulated in a ligand-dependent manner. In fact, while we previously demonstrated the chemical shift changes of the resonances from M215 and M279 located on the TM core bound to the full agonist upon the β arr1 interaction (Shiraishi *et al.*, *Nat. Commun.* (2018) 9, 194, Figure 5), the resonances from the TM core bound to the inverse agonist did not exhibit significant chemical shift changes upon the addition of β arr1 (Supplementary Fig. 4).

Shiraishi *et al.*, *Nat. Commun.* (2018) 9, 194, Fig. 5 Conformation of the TM region of the phosphorylated β_2 AR bound to β -arrestin. Overlay of the 1 H- 13 C HMQC spectra of [2 H-9AA, $\alpha\beta\gamma$ - 2 H, methyl- 13 C-Met] β_2 AR in rHDLs in the

unphosphorylated state (black), phosphorylated state (red), and β -arrestin 1-bound state (blue). The centers of the resonances from M215^{5,54} to M279^{6,41} are indicated with dots. In the phosphorylated state, the resonance from M279^{6,41} could not be observed.

Supplementary Figure 4 | Effect of β arr1 binding on the conformation of the transmembrane region of phosphorylated β_2V_2R bound to an inverse agonist. (A) Distribution of the NMR probes on the structure of the β_2AR TM region. The crystal structure of β_2AR bound to an inverse agonist, carazolol, (PDB ID: 2RH1) [3], is shown as a ribbon model, and the methionine residues observed in the experiments are depicted by sticks. M215 and M279, with resonances that reportedly exhibit significant chemical shift changes upon β arr1 binding in the full agonist-bound state, are highlighted [4]. (B) Overlay of 1H - ^{13}C HMQC spectra of [2H -9AA, $\alpha\beta\gamma$ - 2H , ϵ - ^{13}C -Met] phosphorylated β_2V_2R bound to the inverse agonist (black) and in the complex with β arr1 (red). (C–E) 1H -1D cross-sections of resonances from M215 and M279. The intensity reductions are probably due to the increased apparent molecular weight upon β arr1 binding.

In the revised manuscript, we modified our description of the ligand-dependent interaction to point out the possibility that the C tail could contribute to the ligand-dependent modulation of the β arr1- β_2V_2R interaction, as follows.

(Pages 7-8, lines 112-121)

“Since the same phosphorylation states of full agonist- and inverse agonist-bound β_2V_2R in rHDLs were used, the observed affinity variance for β arr1 is strictly due to differences

in the ligands bound to the TM core. Furthermore, while the resonances from M215 and M279, located on the intracellular side of the TM core, reportedly exhibited significant chemical shift changes upon β arr1 binding in the full agonist-bound state [26], such changes were not observed upon β arr1 binding in the inverse agonist-bound state (Supplementary Fig. 4), suggesting that β arr1 binding to the TM core is modulated by the ligand efficacy. Therefore, the possibility remains that the C tail undergoes conformational changes in a ligand-dependent manner and subsequently affects the interaction with β arr1, it is more likely that β arr1 binds to the TM core of β_2V_2R in rHDLs in a ligand efficacy-dependent manner. Accordingly, β arr1 and β_2V_2R in rHDLs retain the biphasic interaction mode.”

<Comment 3-2>

2. In the second part of RESULTS, “Conformational change of β -arrestin 1 induced by both TM core and C tail interactions”, the authors claim that β -arrestin 1 does not interact with lipid (line 138-141). Is there any control experiment performed to monitor the interaction between β -arrestin 1 and empty lipid? Is there the possibility that in the present experiment the β -arrestin 1 was saturated already and did not observe the interaction between them when adding extra lipid?

We appreciate the reviewer’s question about the effects of lipids on the conformation of β arr1. We have investigated the interaction between β arr1 and empty rHDLs, which did not contain the phosphorylated β_2V_2R , as shown in Supplementary Fig. 6. Upon the addition of empty rHDLs, the resonances from β arr1 did not exhibit significant chemical shift changes. This result indicated that lipids alone did not induce the conformational change of β arr1.

On the other hand, previous fluorescence spectroscopy and cryo-EM studies revealed that arrestins in complex with GPCRs interact with lipids (Lally *et al.*, *Nat. Commun.* (2017) 8:14258, Staus *et al.*, *Nature* (2020) 579, 297-302). Combined with these previous insights, our result shown in Supplementary Fig. 6 suggested that the β arr1-lipid interaction is rather weak, and prior interactions with receptor components are required to observe the β arr1-lipid interaction.

Supplementary Figure 6 | Effect of the addition of rHDLs without receptors on the β arr1 conformation. Overlay of ^1H - ^{13}C HMQC spectra of [u - ^2H , Ile δ 1- $^{13}\text{C}^1\text{H}_3$] β arr1 in the basal state (black) and in the presence of an excess amount of empty rHDLs (red).

In the revised manuscript, we modified our description of the experiment shown in Supplementary Fig. 6 to clarify its purpose, as follows.

(Page 9, lines 147-152)

“To determine whether this conformational change was induced by lipids alone, the interactions between β arr1 and empty rHDLs, lacking phosphorylated $\beta_2\text{V}_2\text{R}$, were investigated. No spectral change was detected upon the addition of excess amounts of empty rHDLs to β arr1 (Supplementary Fig. 6). These results suggest that the conformational change of β arr1 is not induced by interactions with lipids alone, but by those with phosphorylated $\beta_2\text{V}_2\text{R}$ in rHDLs.”

<Comment 3-3>

3. For the conformational analysis by NMR, the methyl group of I377 of β -arrestin 1 which locates at disordered C tail, showed a narrower peak when there is no receptors. However, did this peak disappeared after β -arrestin 1 interacts with $\beta_2\text{V}_2\text{R}$? Since the authors indicated that the C tail region is released from the N-domain of β -arrestin 1 upon interact with the GPCRs and another residue I386 at this region could be observed. The authors should have an explanation about this observation.

We appreciate the reviewer’s question about the resonance from I377 in

the phosphorylated β_2V_2R -bound state. In fact, both resonances from I377 and I386 of β arr1 were observed in the phosphorylated β_2V_2R -bound state, and they were overlapped in the spectra shown in Fig. 2B and Fig. 3A. We can resolve these resonances by processing the spectra without multiplying the apodization function in the 1H dimension, as shown in Supplementary Fig. 7

Supplementary Figure 7 | Magnified view of the resonances from I377 and I386 of β arr1 in the complex with phosphorylated β_2V_2R in rHDLs bound to the full agonist. No apodization function was multiplied to the 1H dimension prior to the Fourier transformation. The projection is displayed above the spectrum.

Both I377 and I386 have exceptionally narrower resonances than the other residues, in good agreement with the fact that the C terminal region of β arr1, including I377 and I386, is highly flexible in the phosphorylated β_2V_2R -bound state.

In the revised manuscript, we added Supplementary Fig. 7 and modified the description of the I377 and I386 resonances in the phosphorylated β_2V_2R -bound state to clarify that both resonances from I377 and I386 were observed with exceptionally narrower linewidths, as follows.

(Pages 10-11, lines 162-169)

“Furthermore, the $\delta 1$ methyl groups of I377 and I386, within the C-terminal region, had exceptionally narrower resonances than other methyl groups in the complex with phosphorylated β_2V_2R in rHDLs bound to the full agonist, indicating that this region is highly flexible (Fig. 2B; middle and Supplementary Fig. 7). Particularly, the resonance from I386 in the complex with phosphorylated β_2V_2R in rHDLs bound to the full agonist was narrower than that in the basal state, indicating that the region including I386 becomes highly flexible upon this interaction (Fig. 2B; middle). This agrees well with the fact that the C terminal region of $\betaarr1$ is released from the N-domain of $\betaarr1$ upon interactions with phosphorylated GPCRs (Fig. 2A) [20, 29].”

REVIEWER COMMENTS

Reviewer #1 (Remarks to the Author):

The authors have addressed all of the raised questions in a thorough and satisfactory way. This reviewer therefore recommends the manuscript be accepted for publication.

Reviewer #2 (Remarks to the Author):

The authors have adequately addressed my suggestions and concerns in their revised manuscript. I can therefore recommend the revised manuscript for publication.

Reviewer #3 (Remarks to the Author):

The authors have addressed all my concerns and I do not have further questions.